# Reconstitution of surface lipoprotein translocation through the Slam translocon

Minh Sang Huynh[1], Yogesh Hooda[1,2], Yuzi Raina Li[1], Maciej Jagielnicki[1], Christine Chieh-Lin Lai[1], Trevor F Moraes[1]*

[1]Department of Biochemistry, University of Toronto, Toronto, Canada; [2]MRC Laboratory of Molecular Biology, University of Cambridge, Cambridge, United Kingdom

**Abstract** Surface lipoproteins (SLPs) are peripherally attached to the outer leaflet of the outer membrane in many Gram-negative bacteria, playing significant roles in nutrient acquisition and immune evasion in the host. While the factors that are involved in the synthesis and delivery of SLPs in the inner membrane are well characterized, the molecular machinery required for the movement of SLPs to the surface are still not fully elucidated. In this study, we investigated the translocation of a SLP TbpB through a Slam1-dependent pathway. Using purified components, we developed an in vitro translocation assay where unfolded TbpB is transported through Slam1-containing proteoliposomes, confirming Slam1 as an outer membrane translocon. While looking to identify factors to increase translocation efficiency, we discovered the periplasmic chaperone Skp interacted with TbpB in the periplasm of *Escherichia coli*. The presence of Skp was found to increase the translocation efficiency of TbpB in the reconstituted translocation assays. A knockout of Skp in *Neisseria meningitidis* revealed that Skp is essential for functional translocation of TbpB to the bacterial surface. Taken together, we propose a pathway for surface destined lipoproteins, where Skp acts as a holdase for Slam-mediated TbpB translocation across the outer membrane.

*For correspondence:
trevor.moraes@utoronto.ca

## Editor's evaluation

This work elucidates the pathway of how surface-exposed lipoproteins of Gram-negative bacteria reach their destination in the outer membrane. Authors have identified an outer membrane protein complex that serves as a translocon for the lipoproteins and discovered the pivotal role of a periplasmic chaperone in the targeting pathway. This work will provide new insights into host invasion mechanisms by pathogenic bacteria, in which surface lipoproteins are critically involved.

## Introduction

Transport of proteins to their correct spatiotemporal location is imperative for cell survival. This key process often requires the movement of proteins across lipid bilayers through a translocation channel which is also referred to as a translocon (*Walter and Lingappa, 1986*; *Schnell and Hebert, 2003*). Translocons are found in all living organisms and include the Sec translocon that is responsible for the bulk of protein transport across the inner plasma membrane (in prokaryotes) and the endoplasmic reticulum membrane (in eukaryotes) (*Johnson and van Waes, 1999*; *Tsirigotaki et al., 2017*). Gram-negative bacteria contain an additional outer membrane that is separated from the plasma membrane (or inner membrane – IM) by the periplasm and a peptidoglycan layer. A number of outer membrane translocons have been previously identified and use different molecular mechanisms to export proteins to the extracellular matrix (*Karuppiah et al., 2011*).

Surface lipoproteins (SLPs) are peripheral membrane proteins that are anchored to the surface of Gram-negative bacteria. These proteins play critical roles in bacterial physiology and virulence (*Wilson and Bernstein, 2016*). Many SLPs were shown to improve bacterial fitness and survival in the host environment, especially for pathogenic bacteria such as *Neisseria*, *Bacteroides*, and *Spirochetes* (*Hooda and Moraes, 2018*). SLPs contain an N-terminal signal peptide that allows their translocation across the inner membrane by the Sec or Tat machinery. In the periplasmic space, the SLPs are modified by three biosynthetic enzymes that cleave the signal peptide and add a lipid group to the N-terminal cysteine residue which anchors them to the inner membrane (*Zückert, 2014*). Most of these lipidated SLPs are recognized by the Lol system, which then delivers SLPs across the periplasm to the inner leaflet of the outer membrane (*Szewczyk and Collet, 2016*; *Okuda and Tokuda, 2011*). In the outer membrane, the protein machinery responsible for the translocation of SLPs across the outer membrane is known only for a handful of SLPs. Recently, an outer membrane protein named Slam (**S**urface **l**ipoprotein **a**ssembly **m**odulator) was identified in the human pathogen *Neisseria meningitidis* that is necessary for the surface display of the SLP transferrin binding protein B or TbpB (*Hooda et al., 2016*). Slam-like proteins were subsequently found in several Gram-negative bacteria from the phylum *Proteobacteria* (*Hooda et al., 2017b*) and has been designated as a type XI secretion system (*Grossman et al., 2021*). Coexpression of Slam1 (the first Slam discovered) with TbpB in the model organism *Escherichia coli*, which lacks both Slam1 and TbpB genes, allows for functional surface display of TbpB. Further, Slam1 was found to interact with TbpB in the outer membrane through coimmunoprecipitation experiments. Taken together, these results confirmed that Slam1 plays a critical role in the transport of TbpB across the outer membrane (*Hooda et al., 2016*). However, the genetic experiments in *N. meningitidis* and the heterologous expression experiments in *E. coli* did not yield a concrete answer as to the exact role of Slam during SLP translocation.

In this study, we developed an in vitro functional assay that allowed us to investigate the role of Slam1 in TbpB translocation. Such assays have been previously developed to study the role of outer membrane protein translocons such as the Bam complex (*Hagan et al., 2010*; *Hagan et al., 2011*), the autotransporter EspP (*Roman-Hernandez et al., 2014*), the two-partner secretion system protein B (*Norell et al., 2014*; *Fan et al., 2012*) and the lipopolysaccharide translocon LptD (*Sherman et al., 2018*). By reconstituting Slam1-mediated TbpB translocation in vitro, we demonstrate that Slam1 acts as an autonomous translocon for the movement of TbpB across the membrane. Using in vitro translocation assays developed in this study, we reveal the first mechanistic insight of Slam-dependent translocation of unfolded lipoproteins across the lipid bilayer. Furthermore, we discovered that the periplasmic chaperone, Skp, interacts with TbpB and another Slam-dependent SLP, named HpuA in the periplasm. We also found that the presence of Skp is crucial for the efficient translocation of TbpB to the surface of *N. meningitidis*. Taken together, we propose a pathway for the localization of SLPs from the cytoplasm to the surface of Slam containing Gram-negative bacteria.

## Results

### Incorporation of Slam1 into liposomes for functional study

Although Slam1 was first discovered in *Neisseria* species (*Hooda et al., 2016*), expressing *Neisseria* Slam1 in *E. coli* for purification and structural study has been proven a challenge in terms of yield and protein stability. Fortunately, our bioinformatic study later revealed hundreds of homologs of *Neisseria* Slam1 in other Gram-negative bacteria (*Hooda et al., 2017a*) including the *Moraxella catarrhalis* Slam1 (Mcat Slam1) that has 40.3% sequence identity. The purification of Mcat Slam1 in n-dodecyl-B-D-maltoside (DDM) proved to be more stable and resulted in much higher yields (*Figure 1—figure supplement 1*).

To evaluate the feasibility of characterizing the function of *M. catarrhalis* Slam1 (Mcat Slam1) in SLP translocation, we coexpressed Mcat Slam1 with its substrate Mcat TbpB and examined the display of Mcat TbpB on the surface of *E. coli* (*Figure 1—figure supplement 2a*). The results showed that TbpB was successfully reconstituted on the surface of *E. coli* cells when it was coexpressed with either *N. meningitidis* Slam1 or *M. catarrhalis* Slam1 (*Figure 1—figure supplement 2b*). Coexpression of TbpB with the negative control Slam2, a homolog of Slam in *N. meningitidis* and *N. gonorrhoeae* that are responsible for the surface display of another SLP, hemoglobin–haptoglobin utilization protein A (HpuA) (*Hooda et al., 2017a*), failed to reconstitute TbpB on the surface of *E. coli*. This experiment

illustrated that TbpB is specifically reconstituted on the bacterial cell surface by Slam1 and suggested that the components from *E. coli* are sufficient for Slam1-dependent TbpB translocation. Thus, *M. catarrhalis* Slam1 was used as a model for functional studies in this paper.

To determine whether Slam1 is an outer membrane translocon working independently from other major translocation systems such as the Bam complex, we attempted to reconstitute the Slam1-dependent TbpB translocation with minimal components. First, we tested the incorporation of purified Mcat Slam1–DDM complex (N-terminal his-tag) into liposomes. Detergent removal allowed for successful insertion of Mcat Slam1 as seen by sodium dodecyl sulfate–polyacrylamide gel electrophoresis (SDS–PAGE), and western blot analysis using α-his antibodies (*Figure 1—figure supplement 3*). To examine liposome insertion, we used the *E. coli* BamABCDE complex as a control. BamABCDE was purified as previously described (*Hagan et al., 2011*) and could potentiate the insertion of the outer membrane protein OmpA into liposomes (*Figure 1—figure supplement 4*). Insertion of Mcat Slam1 or Bam complex into liposomes did not affect the stability of liposomes as proteoliposomes containing these proteins were able to float to the top of sucrose gradients upon ultracentrifugation (*Figure 1—figure supplement 5a*). Further, to examine the orientation of Mcat Slam1 and Bam complex in proteoliposomes, we incubated the Mcat Slam1 and BamABCDE containing proteoliposomes with proteinase K. The addition of proteinase K led to formation of low-molecular weights bands in an SDS–PAGE gel (marked with asterisk, *Figure 1—figure supplement 5b*, left panel) and loss of Slam1 band in an α-His western blot (*Figure 1—figure supplement 5b*, right panel), indicating that over 80% of Slam is inserted with its periplasmic domain protruding from the surface – the 'inside-out' orientation required for the in vitro translocation assay.

## Slam1 proteoliposomes translocate purified unfolded substrate

Once we established a proteoliposome with Slam1 incorporation, we attempted to detect the Slam-mediated transport of SLPs across the bilayer (*Hagan et al., 2010*). To this end, we purified lipidated functional *M. catarrhalis* TbpB in DDM detergent for the assay (*Figure 1—figure supplement 6*) and then unfolded the proteins using urea. DDM was removed by SM2 biobeads, and the urea concentration was reduced upon diluting the urea-unfolded TbpB into the preformed proteoliposomes (*Figure 1a*). Translocation of TbpB was assessed by sensitivity to proteinase K. Only the urea-unfolded TbpB was successfully translocated into Slam1 proteoliposomes (~3% protection), but not in empty liposomes or Bam proteoliposomes (*Figure 1b*). Although the protection is low, the analysis of variance (ANOVA) test determined the translocation efficiency is significant between Slam1 proteoliposomes and the two negative controls for unfolded TbpB (*Figure 1c*). This protection results from translocation of TbpB into the lumen of the Slam containing proteoliposome and not from Slam1–TbpB interactions, as a proteinase K digestion of TbpB in the presence of Slam1–DDM results in completely digested TbpB in the solution (*Figure 1—figure supplement 7* – upper panel). Moreover, the protection of TbpB was specific to Slam1 proteoliposomes only, as Slam1 proteoliposomes did not protect the negative control AfuA, a nonlipoprotein that typically resides in the periplasm of *Actinobacillus pleuropneumonia* (*Sit et al., 2015*; *Figure 1—figure supplement 7f* – lower panel). The addition of the Bam complex did not affect the TbpB translocation efficiency, suggesting that Bam complex does not play a role in this translocation process. Finally, the low efficiency of insertion observed for the defined system, together with the observation that translocation across Slam1 proteoliposomes occurs only when the TbpB is denatured by urea, lead us to hypothesize that there are likely additional periplasmic factors that keep the SLP unfolded for an efficient translocation.

## Translocation of TbpB via Slam1 requires periplasmic components but the process is independent from the release of TbpB from the inner membrane

To delve deeper into the mechanism of Slam-mediated SLP translocation and whether additional of periplasmic contents are required for efficient translocation, we examined the translocation of TbpB presented by *E. coli* spheroplasts that lack an intact outer membrane (*Norell et al., 2014*; *Figure 2a*). Like other lipoproteins, after being expressed and translocated across the inner membrane into the periplasm, TbpB is modified by Lgt, Lsp, and Lnt like other lipoproteins and displayed on the outer surface of the inner membrane (*Hooda et al., 2016*). Previous studies have shown that the addition of the periplasmic chaperone LolA leads to release of SLPs from spheroplasts into the culture

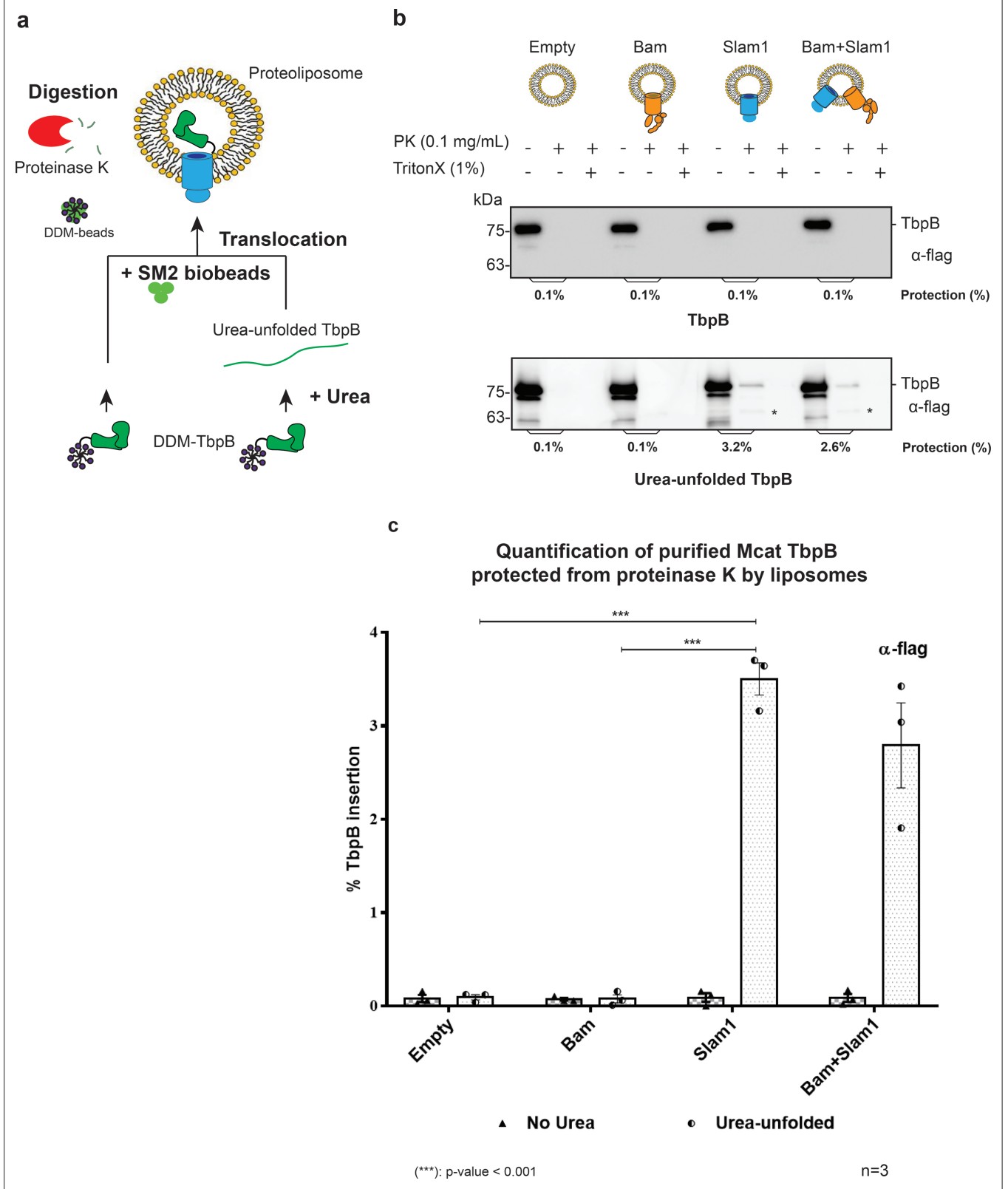

**Figure 1.** Slam1 is necessary for translocation of unfolded TbpB. (**a**) Model of a defined in vitro assay for TbpB translocation. *M. catarrhalis* TbpB (folded and urea-unfolded) is translocated inside Slam1-containing proteoliposomes. SM2 biobeads were used to remove DDM detergent from TbpB before adding proteoliposomes for translocation. Efficiency of TbpB translocation/insertion was calculated based on percentage of TbpB that was protected from proteinase K. (**b**) Representative proteinase K protection assay results obtained for Slam1 or Slam1 + Bam incubated with purified TbpB (folded

*Figure 1 continued on next page*

*Figure 1 continued*

or kept unfolded by 8 M urea). Proteoliposomes containing Empty or Bam were used as controls. Each sample was treated with PK or PK + Triton X-100 and examined by western blot. α-flag antibody western blots were used to quantify the amount of TbpB. Asterisk (*) The lower band in the (+PK and −Triton X) samples in Slam1 and Bam + Slam1 proteoliposomes treatment likely represents incompletely translocated TbpB that has been partially degraded by proteinase K. (**c**) Quantification of TbpB protection in proteoliposomes through densitometry analysis. The % TbpB insertion was calculated by dividing the protected TbpB of + PK sample by TbpB of the input sample. The plot contains results obtained from three biological replicates. Individual data points were included on the graph. Two-way analysis of variance (ANOVA) test was performed to determine the statistical significance for the translocation of unfolded TbpB by Slam1 proteoliposomes and Slam1 + Bam proteoliposomes treatment versus by the negative controls (empty liposomes and Bam proteoliposomes). Only statistical significance of the unfolded TbpB translocation by Slam1 proteoliposomes against the two negative controls are included on the blot for simplification. (***) represents $p$-value < 0.001.

The online version of this article includes the following source data and figure supplement(s) for figure 1:

**Source data 1.** Quantification of in vitro translocation of purified TbpB.

**Figure supplement 1.** Purification of *M. catarrhalis* Slam1.

**Figure supplement 2.** Translocation of Mcat TbpB to the surface of *E. coli* cells.

**Figure supplement 2—source data 1.** Fluorescent signal of TbpB on the surface of E.coli.

**Figure supplement 3.** Generation of Slam1 and Bam proteoliposomes.

**Figure supplement 4.** Purification and characterization of *E. coli* Bam complex.

**Figure supplement 5.** Characterization of Mcat Slam1 and BamABCDE containing proteoliposomes.

**Figure supplement 6.** Purification and functional characterization of *M. catarrhalis* TbpB.

**Figure supplement 7.** Protection of urea-unfolded TbpB (flag-tag) and negative control urea-unfolded AfuA (his-tag) by liposomes, proteoliposomes, and purified Slam1–DDM complex.

supernatant (*Tajima et al., 1998*). Hence, we purified *E. coli* LolA and tested LolA-dependent release of Mcat TbpB from spheroplasts (*Figure 2—figure supplement 1*). Higher amounts of TbpB were detected in the supernatant in the presence of LolA. We incubated the TbpB expressing spheroplasts directly with Slam1 or Bam proteoliposomes and estimated the translocation efficiency of TbpB using a proteinase K assay (spheroplast-dependent translocation). Any TbpB translocated into the lumen of the proteoliposome should be protected from proteinase K digestion. From this assay, we found that proteoliposomes containing Slam1 showed significantly higher protection (40%) compared to Bam proteoliposomes or empty liposomes (5%) (*Figure 2b* – upper panel). The protection of TbpB was lost upon the addition of Triton X-100 suggesting that TbpB is shielded from the protease activity by the lipid bilayer of the liposomes. The background protection observed in empty and Bam proteoliposomes is inherent to the procedure as experiment without the presence of liposome showed similar background protection, suggesting that the background protection may originate from the spheroplast secretions themselves (*Figure 2—figure supplement 2a*). Interestingly, proteoliposomes containing both Bam complex and Slam once again did not improve the efficiency, indicating that the translocation of TbpB does not require Bam complex.

The success of the in vitro Slam-dependent translocation of spheroplast-released SLPs into liposomes provided an assay to investigate SLP translocation in greater detail. Many OMPs require inner membrane factors for energy transduction such as TonB-dependent receptors (*Pawelek et al., 2006*) or chaperone activity TamA (*Stubenrauch et al., 2016*) to perform their function. Studies of the Lol system have shown that LolA shuttles between the inner membrane and the outer membrane (*Szewczyk and Collet, 2016*), and hence we predicted that Slam-mediated SLP translocation does not require any inner membrane factors unlike the Lpt system (*Sherman et al., 2018*). To validate this hypothesis, we incubated the empty, Bam, Slam1, or Bam + Slam1 proteoliposomes with the supernatant isolated from spheroplasts that were expressing TbpB (spheroplast-independent translocation). As seen previously in the spheroplast-dependent translocation assay, we observed similar TbpB protection from proteinase K in proteoliposomes containing Slam1 (~40% protection) and Bam + Slam1 (~35%) (*Figure 2b* – lower panel) but not empty (~7%) nor Bam (~5%) proteoliposomes. Interestingly, we did not observe any loss in translocation efficiency between spheroplast-dependent and -independent assay (*Figure 2c*), confirming that Slam-mediated SLP translocation is independent of SLP release from the inner membrane. This differs from other secretion systems that require partners in the inner membrane who provide energy through ATP/proton motive force (*Sherman et al., 2018*;

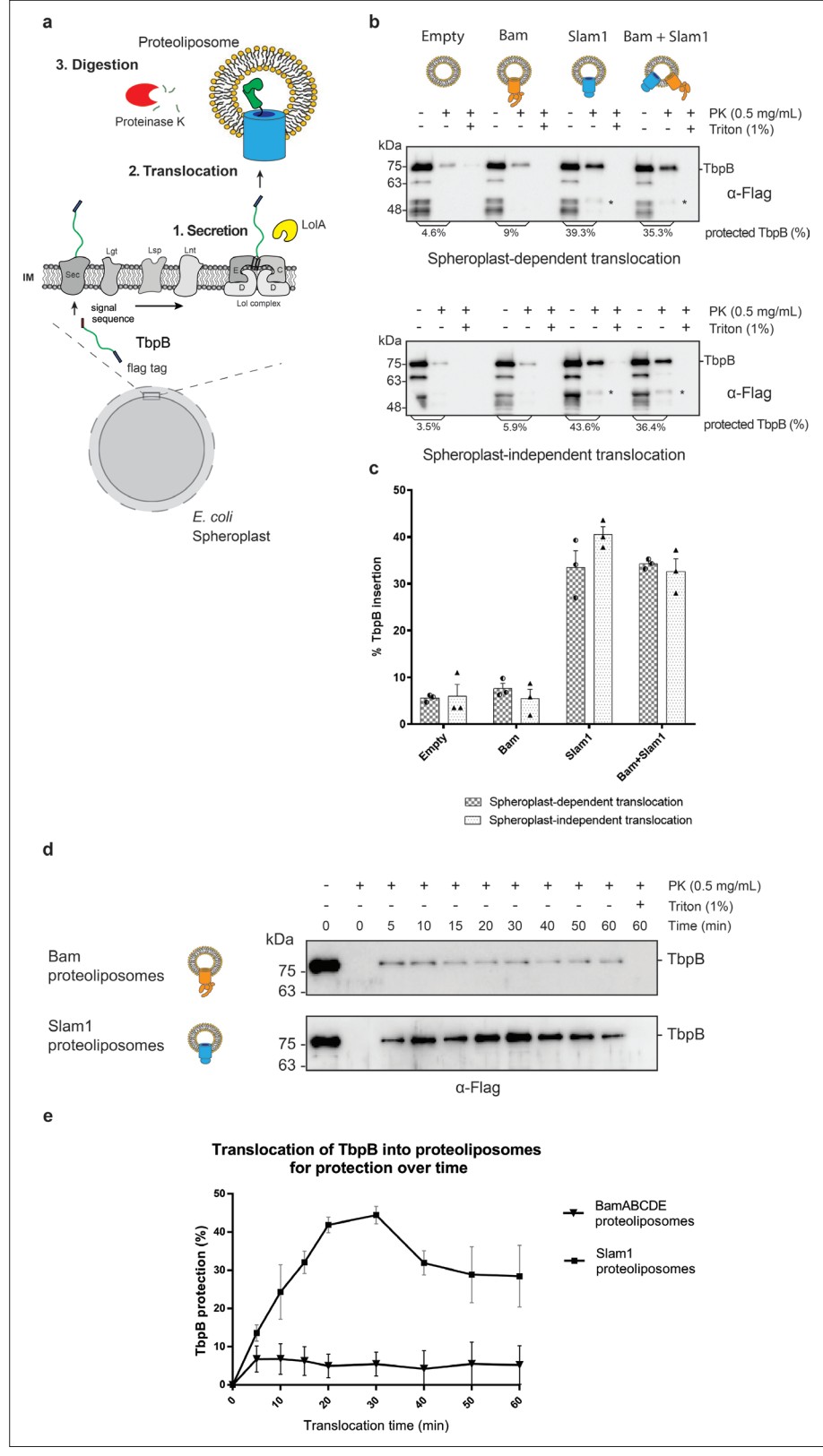

**Figure 2.** In vitro translocation assay for reconstitution of Slam-dependent SLP translocation. (**a**) Model of the proposed in vitro proteoliposomes translocation assay for TbpB secreted directly from *E. coli* spheroplast. As *E. coli* cell expressing TbpB, the cells were converted into spheroplast that has an intact inner membrane. Purified LolA was added to release mature TbpB (processed by Lgt, Lsp, and Lnt) from the LolCDE complex in

*Figure 2 continued on next page*

*Figure 2 continued*

the inner membrane. The secreted TbpB was incubated with proteoliposomes for translocation, followed by PK digestion to quantify the amount of TbpB that had been translocated inside the liposomes. (**b**) Representative α-flag western blots obtained for the in vitro translocation assay. Slam1 proteoliposomes were incubated either with spheroplasts expressing TbpB (spheroplast-dependent translocation, upper panel) or supernatant of spheroplasts that have been induced for TbpB production (spheroplast-independent translocation, lower panel). Empty liposomes and Bam proteoliposomes were used as controls. Proteoliposomes containing Bam + Slam1 were used to test if the Bam complex plays an accessory role to Slam in TbpB translocation. For each proteoliposome, no proteinase K treatment (−PK), proteinase K treatment (+PK) and proteinase K + Triton X-100 treatment (+PK + T) samples are shown. The % TbpB protection shown was calculated by dividing the intensity of the mature TbpB band (~75 kDa) for each sample by the −PK sample. (*) Partial TbpB fragment which is only seen in the presence of Slam1 proteoliposomes. (**c**) Quantification of TbpB protection in proteoliposomes through densitometry analysis. The plot represents data obtained from at least three biological replicates for both spheroplast-dependent translocation and spheroplast-independent assay. Individual data points were included on the graph. (**d**) Representative α-flag western blot of spheroplast-independent TbpB translocation into Bam and Slam1 proteoliposomes over time. Spheroplast-secreted TbpB was incubated with proteoliposomes in 1:1 ratio at room temperature. Samples were collected every 5 or 10 min and left on ice before proteinase K treatment. (**e**) Quantification of spheroplast-secreted TbpB translocation into Bam proteoliposomes and Slam1 proteoliposomes over the course of 60 min.

The online version of this article includes the following source data and figure supplement(s) for figure 2:

**Source data 1.** Quantification of in vitro translocation of spheroplast-secreted TbpB.

**Source data 2.** Quantification of time-dependent translocation of spheroplast-secreted TbpB.

**Figure supplement 1.** Purification and functional characterization of LolA from *E. coli*.

**Figure supplement 2.** Characterization of spheroplast-secreted TbpB.

---

*Stubenrauch et al., 2016*). This finding suggests Slam-dependent SLP translocation is akin to two-partner secretion systems (*Fan et al., 2012*; *Norell et al., 2014*; *Guérin et al., 2017*).

To further examine the fate of TbpB after being translocated into Slam1 proteoliposomes, samples of proteinase K protected solutions were treated with 1 mM PMSF to inhibit proteinase K activity before being dissolved in 0.1% Triton X and incubated with iron-loaded human transferrin-conjugated beads (*Figure 2—figure supplement 2b*). Although the amount of the protected sample from Bam and Slam1 proteoliposomes differs (5% and 43%, respectively), none of the TbpB protected by the negative control Bam proteoliposomes binds to human transferrin beads. Meanwhile, the TbpB that is translocated and protected by Slam1 proteoliposomes was shown to retain its functionality through binding to human transferrin-conjugated resin. This result suggested that the TbpB which is protected by Slam1 proteoliposomes is properly folded after being translocated inside the liposomes.

## Translocation of spheroplast-secreted TbpB into Slam1 proteoliposomes is time dependent

To further elaborate the function of Slam1 in the translocation of TbpB into proteoliposomes, we performed a time-dependent translocation assay over the course of 60 min. Samples were withdrawn every 5–10 min and incubated on ice before being treated with proteinase K. The result showed that Slam1 translocation for TbpB reaches the maximal efficiency (40–43%) between 20 and 30 min (*Figure 2d, e*). Compared with other major translocons such as the Bam complex (10–20 min) (*Hussain and Bernstein, 2018*), Slam1-dependent translocation takes a similar amount of time to reach maximal translocational activity (*Figure 1—figure supplement 4c*). Notably, we also observed the protection of TbpB reduced after 30 min but this could be attributed to the instability of proteoliposomes after a period of time with changes in pH, protein composition, size, and charge of macromolecules (*Nakhaei et al., 2021*).

## Periplasmic chaperone Skp interacts with prefolded TbpB in the periplasm

As mentioned above, the Slam1-dependent translocation requires TbpB to be unfolded and hence, we hypothesized that other factors in the periplasm bind the SLPs and prevent their premature folding prior to Slam-mediated translocation. To identify periplasmic factors that might be involved in the

translocation, periplasmic TbpB complexes were isolated using an affinity flag-tag on its C-terminus (*Figure 3a*). The pulldown fraction was analyzed using mass spectrometry and western blots. In this pulldown assay, AfuA – a well-folded periplasmic nonlipoprotein from *A. pleuropneumonia* was used as a negative control to rule out nonspecific periplasmic protein interactions (*Sit et al., 2015*). Skp – a periplasmic chaperone was the only protein that was identified in the pulldown of TbpB but not in the negative control (*Table 1*). The mass spectrometry results were validated using western blots (*Figure 3b*). As expected, LolA only interacts with lipoprotein TbpB in the elution fraction (*Figure 3b* – middle panel). In addition, the blot confirmed the presence of periplasmic protein in the elution of TbpB (*Figure 3b* – bottom panel).

To further validate the interaction between Skp and SLPs, a reciprocal pulldown assay was performed in which a purified His-tagged chaperone was added into the spheroplast prior to the secretion of SLPs (*Figure 3c*). In this assay, we also examined whether Skp interacts with other SLPs such as hemoglobin–haptoglobin utilization protein (HpuA) – a substrate of Slam2 homolog in *N. meningitidis* (*Hooda et al., 2016*). Two other periplasmic chaperones which are known to be involved in the transport of OMPs, SurA, and DegP were also examined (*Sklar et al., 2007*). As expected, none of the chaperones pulled down the negative control protein AfuA (*Figure 3d* – first panel). Although chaperones SurA and DegP showed no binding to TbpB, a small amount of HpuA was detected in the elution fraction (*Figure 3d* – third panel). However, these interactions were probably nonspecific since the intensity of HpuA bands detected in SurA and DegP elutions are similar to the nonspecific binding levels detected in the negative control Ni-NTA beads (*Figure 3d* – third panel). The coimmunoprecipitation experiments confirmed that only the periplasmic chaperone Skp interacts with the spheroplast-released TbpB and HpuA, as these lipoproteins were found in the elution (E) fraction of the Skp pulldown (*Figure 3d* – second and third panels). Skp is a well-studied homotrimeric chaperone which is known to bind unfolded proteins in the periplasm to prevent their aggregation and degradation (*Walton et al., 2009*; *Volokhina et al., 2011*). Skp has been shown to be important for OMP membrane insertion by the Bam complex (*Sklar et al., 2007*; *Mas et al., 2019*). Our findings suggest that Skp also interacts with TbpB-like SLPs in the periplasm and assists in their translocation across the outer membrane.

## Periplasmic chaperone Skp is essential for Slam-dependent translocation in *E. coli*

To determine whether Skp is essential for the translocation of SLPs via Slam, we co-expressed TbpB and Slam1 in K12 *E. coli* strains devoid of functional Skp or DegP (as a negative control) (*Baba et al., 2006*). The presence of TbpB on the surface of *E. coli* was detected using rabbit α-flag antibody, followed by phycoerythrin-conjugated α-rabbit IgG . The *E. coli* K12 *Δskp* mutant had significant reduction of TbpB surface exposure (50%) compared to wild-type cells (*Figure 4a*). Depletion of DegP slightly reduced the translocation of TbpB (12%) but this was not statistically significant (by one-way ANOVA test). No reduction in the expression of either Slam1 or TbpB was observed in western blots. Furthermore, the processing of TbpB by signal peptidase II (lower band of TbpB ~75 kDa) and subsequence release from the inner membrane was unaffected suggesting the defect in surface display by Skp occurs after the release of TbpB from the inner membrane.

To further investigate the role of periplasmic chaperone Skp, we leveraged our in vitro translocation assay using Slam1 proteoliposomes and spheroplast-secreted TbpB. TbpB that was secreted from K12 spheroplast mutants that lacked Skp or DegP, was incubated with Slam1 proteoliposomes for translocation. The overall results were consistent with the in vivo translocation in K12 *E. coli* (*Figure 4b*). In comparison with wild-type spheroplast TbpB, the *Δskp*-spheroplast-secreted TbpB failed to translocate inside of Slam1 proteoliposomes, while the translocation efficiency of *Δdegp*-spheroplast-secreted TbpB was only marginally reduced. This suggests that Slam-mediated translocation of SLPs requires the periplasmic chaperone Skp.

## Skp increases the translocation of purified TbpB in Slam1-containing proteoliposomes

Given that Skp is necessary for Slam-mediated translocation, we hypothesized that addition of purified *E. coli* Skp should increase the translocation efficiency of purified TbpB into Slam1 proteoliposomes. To test this hypothesis, we purified *E. coli* Skp and LolA and added these to urea-denatured

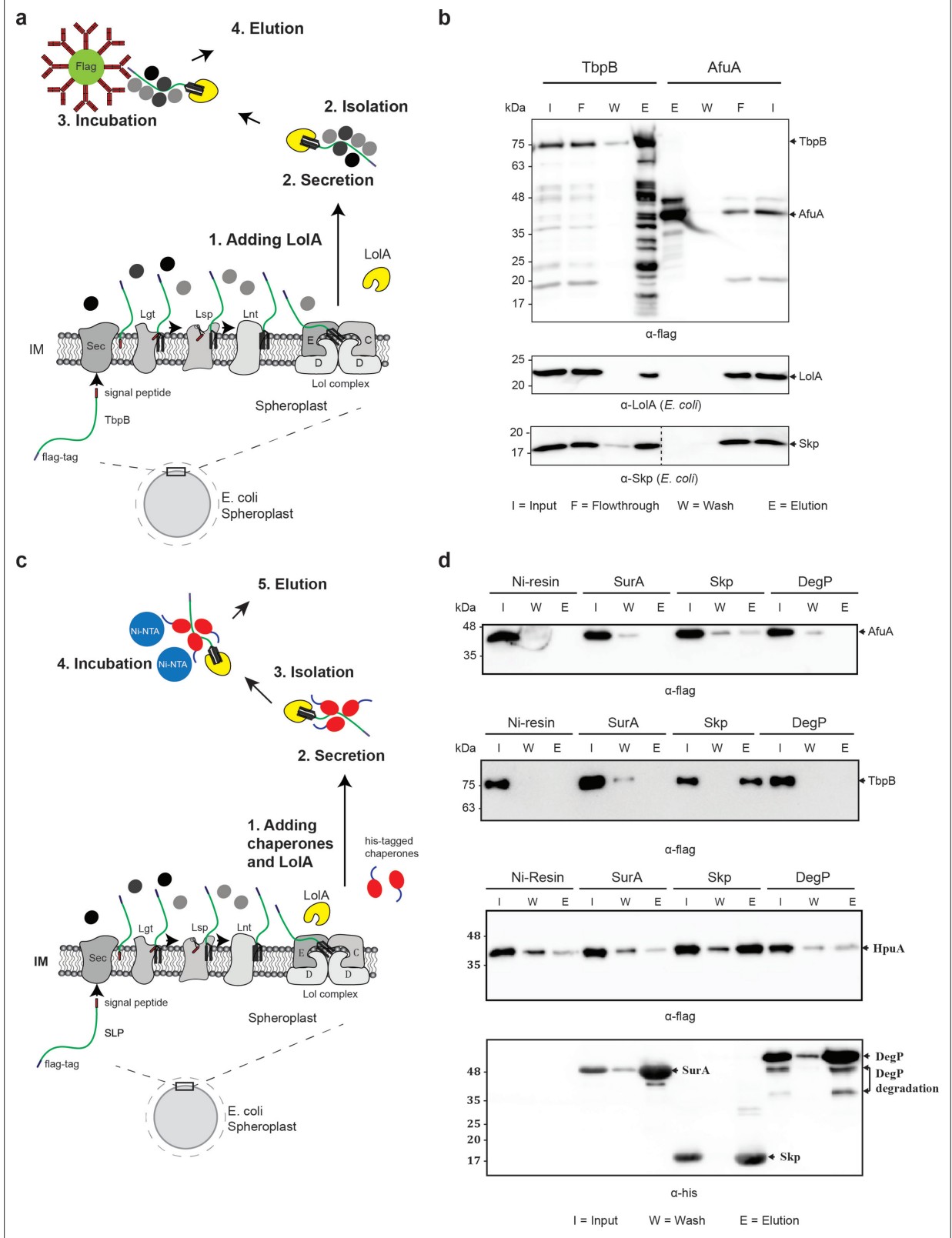

**Figure 3.** Periplasmic chaperone Skp interacts with surface lipoproteins TbpB and HpuA after being released from the inner membrane. (**a**) Model of pulldown assay using the flag-tag on the C-terminus of TbpB. Samples were analyzed using mass spectrometry (summarized in **Table 1**) and examined on western blots. (**b**) Representative western blots for TbpB pulldown assay followed by mass spectrometry. LolA and periplasmic chaperone Skp were detected in the TbpB eluted fraction. No LolA and Skp were eluted in negative control AfuA experiment. Note: Upper and lower panels are from

*Figure 3 continued on next page*

*Figure 3 continued*

the same blot. Bottom panel was run on a second blot and the dashed line in the middle to indicate the exclusion of lanes (bead lanes – not shown) between the control (AfuA) and treatment (TbpB). (**c**) Model of reciprocal pulldown assay using the his-tag on the N-terminus of chaperones. Purified his-tagged chaperones (SurA, Skp, and DegP) were added to the spheroplast before the induced secretion of SLPs. (**d**) Representative western blots of the reciprocal pulldown assay. Only periplasmic chaperone Skp (17 kDa) was found to pull down lipoprotein TbpB and HpuA while no AfuA was found in the Skp elution fraction. All three proteins AfuA (negative control), TbpB and HpuA (Slam-dependent lipoproteins) were not found in the elution fraction of SurA and DegP pulldown. Note: α-his blot was used to detect chaperones and α-flag was used to detect HpuA samples within the same experiment. Elution volumes were 100 µl (1:10 input). DegP has molecular weight at 54 kDa. Lower band is the result of self degradation.

The online version of this article includes the following source data for figure 3:

**Source data 1.** Mass spectrometry of pulldown samples.

TbpB prior to incubation with Slam1-containing proteoliposomes. Periplasmic chaperone SurA was also purified and used as a negative control as we have shown that SurA does not interact with TbpB (*Figure 3d* – second panel). As expected, with the addition of Skp to Slam1 proteoliposomes, the translocation efficiency of TbpB significantly increased by 10-fold (30% protection) in comparison with addition of SurA into the same proteoliposomes (3.5% protection) (*Figure 4c*). Notably, the addition of Skp to the empty liposomes also increased TbpB's protection by 3-fold (up to 10% protection) relative to the control SurA + Slam1 proteoliposomes (*Figure 4c* – right panel). This result is comparable to with the background protection observed for the empty liposomes in the translocation assays which used TbpB secreted directly from spheroplast (*Figure 2b*). The protection might be due to protease resistance that chaperones provide for their substrates in the periplasm which has previously also been reported for unfolded OMPs (*Yan et al., 2019*).

To confirm that the background protection is from the protease resistance of chaperone–substrate complexes, the samples were spun down against a sucrose gradient (0–60% wt/vol) after the proteinase K treatment to isolate only the proteoliposomes. The western blots and Coomassie blue stained SDS–PAGE showed a clear separation of the two components, the intact proteoliposomes in the top layer and unincorporated proteins in the middle and bottom layers of the sucrose gradient (*Figure 4d* – left panel). While most of the unincorporated proteins were located in the bottom and middle fractions, significant amounts of Slam1 and TbpB were found only in the top fractions of Slam1 proteoliposomes (*Figure 4d* – upper left panel). In comparison to the SurA + Slam1 proteoliposomes, the addition of Skp to Slam1 proteoliposomes increased translocation efficiency for TbpB by almost 3-fold (*Figure 4d* – right panel). This ratio supports the previous result of the 3-fold increase in protection from proteinase K observed when comparing the Skp–TbpB chaperone-mediated protection with the translocation-mediated protection observed with Skp–TbpB + Slam1 proteoliposomes (*Figure 4c* – right panel). There was no trace of TbpB in either of the negative controls which could be attributed to the presence of trace amounts of proteinase K which could have continuously degraded the unfolded TbpB during the 18 hr ultracentrifuge run. Taken together, these results suggest that Skp plays an important role in the translocation of TbpB to the surface via Slam1, likely through its holdase function.

**Table 1.** Summary of mass spectrometry for flag-tagged TbpB pulldown.

| | AfuA | | | | TbpB | | | |
|---|---|---|---|---|---|---|---|---|
| | Proteins | Total peptides | % Coverage | Location | Proteins | Total peptides | % Coverage | Location |
| 1 | AfuA | 380 | 78% | P | TbpB | 265 | 31% | – |
| 2 | TufA | 119 | 70% | IM | LolA | 146 | 73% | P |
| 3 | LolA | 36 | 72% | P | TufA | 57 | 59% | IM |
| 4 | OmpF | 12 | 34% | OM | OmpF | 21 | 38% | OM |
| 5 | DegP | 10 | 30% | P | DegP | 15 | 32% | P |
| 6 | FlgH | 12 | 34% | P | OmpA | 15 | 34% | OM |
| 7 | DegQ | 6 | 14% | P | Skp | 9 | 27% | P |
| 8 | OmpA | 5 | 22% | OM | FlgH | 8 | 26% | P |
| 9 | | | | | DegQ | 5 | 13% | P |

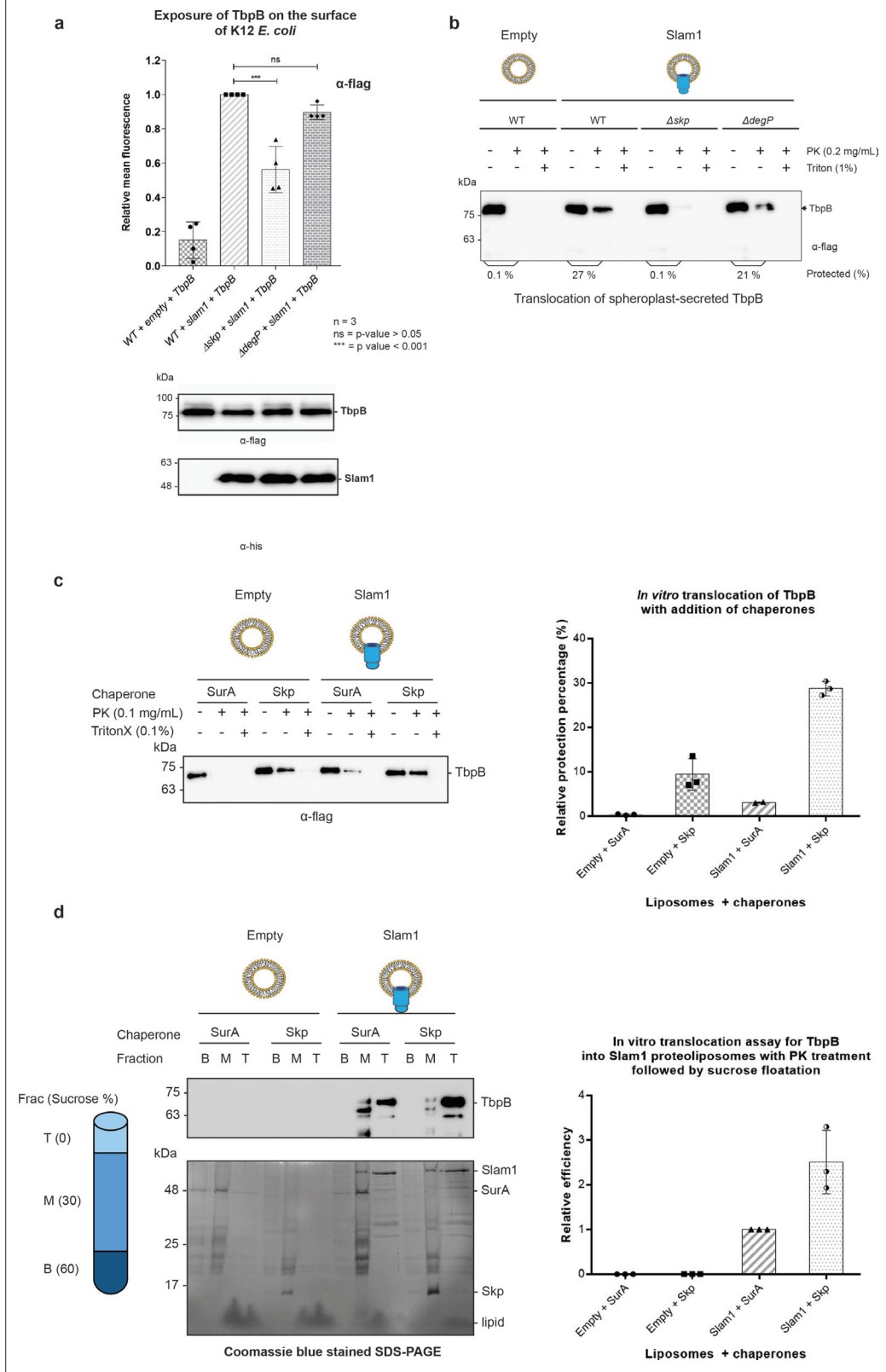

**Figure 4.** Periplasmic chaperone Skp is required for Slam1–TbpB translocation in the reconstitution systems. (**a**) Translocation of TbpB via Slam1 to the surface of *E. coli* K12 mutants. Depletion of Skp significantly reduces the translocation of TbpB to the surface (by 50% – detecting by using α-flag antibody). Bottom panel – western blots of whole cell lysates detecting the expression of TbpB and Slam1 in K12 *E. coli* mutant strains. The processed

*Figure 4 continued*

TbpB (high intensity lower band at 75kDa) was unaffected in the mutant *E. coli* indicating that absence of one of the chaperones did not affect the maturation of TbpB in the periplasm. (***) represents p-value < 0.001 and (ns) represent p-value > 0.05. (**b**) Representative western blot of the in vitro proteoliposomes translocation for TbpB secreted from K12 *E. coli* spheroplast mutants. TbpB secreted from Δ*skp* spheroplast fails to translocate into the Slam1 proteoliposomes for protection against proteinase K. (**c**) Representative western blot (left panel) and quantification (right panel) of the in vitro translocation of purified TbpB into Slam1 proteoliposomes in addition of purified chaperones. Full length lipidated TbpB was unfolded in urea followed by incubation with LolA and either SurA (negative control) or Skp before incubating with empty or Slam1 proteoliposomes and proteinase K digestion. TbpB–Skp complex provided extra protection for TbpB even in the absence of Slam1. (**d**) Representative western blot (left panel) and quantification (right panel) of the protected TbpB by the liposomes which were isolated using sucrose flotation assay after proteinase K digestion. Translocation of TbpB into Slam1 proteoliposomes increased by 2.5-fold in the presence of Skp in comparison with the Slam1 proteoliposomes + SurA (positive control). Results are from at least three biological replicates. Individual data points were included on the graph.

The online version of this article includes the following source data for figure 4:

**Source data 1.** Fluorescence of TbpB on surface of *E. coli*.

**Source data 2.** Quantification of in vitro translocatio.

**Source data 3.** Quantification of in vitro translocation following sucrose floatation assay.

## Deletion of Skp in B16B6 decreases the exposure of TbpB on the surface of *N. meningitidis*

To examine the role of Skp in the Slam-dependent translocation of SLPs in B16B6 *N. meningitidis* that contains endogenous TbpB and Slam1, we deleted the gene *skp* (Δskp strain) and examined its effect on surface display of TbpB (*Figure 5—figure supplements 1 and 2*). Such experiments have been previously done in other studies for periplasmic chaperones SurA, Skp, and DegQ (homolog of DegP) in *N. meningitidis* in which a single deletion of either one of the chaperones did not affect cell vitality nor the expression of OMPs or their insertion into the outer membrane via the Bam complex (*Volokhina et al., 2011*). In our study, the deletion of *skp* (Δskp) overall did not affect the growth of *N. meningitidis* as the cells reached the optimal $OD_{600}$ after 12 hr (*Figure 5—figure supplement 3a*). However, the total amount of TbpB detected after 18 hr of growth was significantly diminished (*Figure 5—figure supplement 3b*). Such a reduction was also observed in a Δ*slam1* strain, and it has been shown in the previous study that in the absence of Slam1, the built-up TbpB in the periplasm was subjected to degradation (*Hooda et al., 2016*). Similar rationale can be made here for the reduction of TbpB in Δ*skp* strain, as without Skp to keep unfolded TbpB for Slam1 translocation, the built-up TbpB was eventually degraded by periplasmic proteases. To examine whether the deletion of Skp only affected the translocation of TbpB to the surface, equal expression of TbpB is also needed for the comparison. Thus, the cells were treated with 0.1 mM deferoxamine, an iron-chelating agent, to induce the expression of proteins that are involved in iron acquisition such as TbpB (*Fegan et al., 2019*). In this assay, we used α-TbpB antibody to probe for TbpB on the cell surface. Unlike the two negative controls (Δ*tbpB* and Δ*slam1*) which completely inhibit the translocation of TbpB, deletion of Skp reduces the amount of TbpB on the surface about 50% comparing to the wild-type strain (*Figure 5a* – top panel). Whole cells samples of these strains were analyzed on western blots to assess the expression of the Slam1–TbpB pair (*Figure 5b*). The results indicated that the amount of TbpB within the mutant strains (Δ*slam1* and Δ*skp*) were successfully induced by deferoxamine to a comparable level of TbpB expression in the wild-type strain (*Figure 5b* – top panel) and depletion of Skp did not affect the expression level of Slam1 (*Figure 5b* – middle panel). Thus, the reduction of TbpB on the surface of the Δ*skp* strain is likely due to the decrease in translocation activity of Slam1. This result is comparable to the translocation of Mcat TbpB to the surface of the *E. coli* Δ*skp* mutant in which the signal from a C-terminal flag-tag was used to assess the surface display of the lipoprotein (*Figure 4c*).

To examine whether these surface exposed TbpB in B16B6 Δ*skp* strain is functional, we probed the cells using biotinylated human transferrin (*Calmettes et al., 2012*). A 5-fold reduction in binding to biotinylated human transferrin was observed for Δ*skp N. meningitidis* strain, indicating a significant loss of functional TbpB assembled on the surface of *N. meningitidis* (*Figure 5a* – bottom panel). The complementation of *skp* from the pGCC4 vector successfully rescued the translocation of TbpB of the

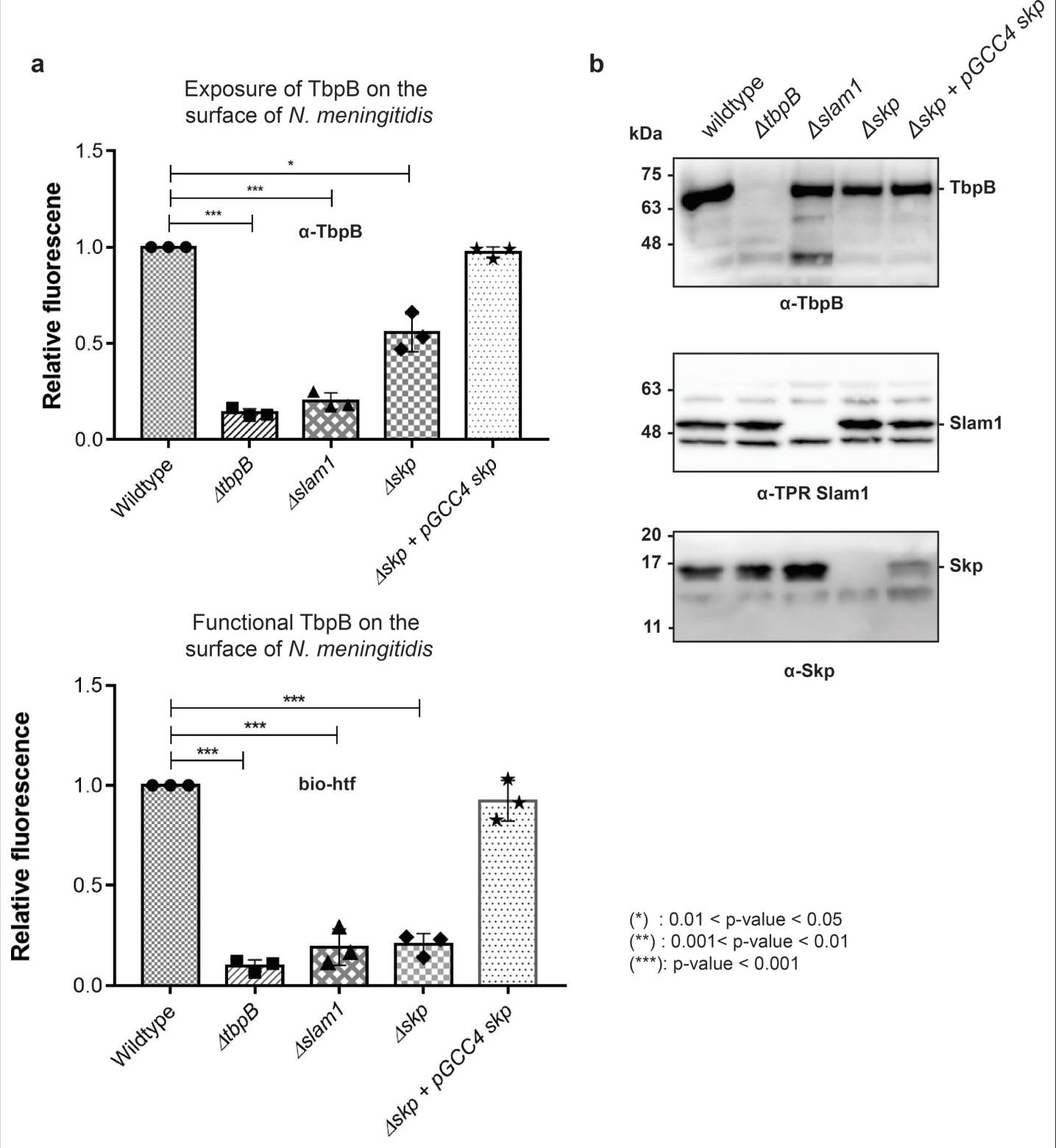

**Figure 5.** Periplasmic chaperone Skp is important for translocation of TbpB to the surface of *N. meningitidis*. (**a**) Relative fluorescence intensity accessing the presence of TbpB (65 kDa) on the surface of *N. meningitidis* mutants using α-TbpB antibody (exposure – top) and biotinylated human transferrin (functional – bottom). Individual data points were included on the graph. Depletion of Skp decreased surface exposed TbpB by 50% and the translocated TbpB is nonfunctional (not binding to biotinylated human transferrin). Complementation of *skp* from pGCC4 vector and 0.1 mM IPTG successfully restored the translocation of TbpB and its function. (*) represent 0.01 < p-value < 0.05 and (***) represent p-value < 0.001. (**b**)

*Figure 5 continued on next page*

*Figure 5 continued*

Representative western blots to access the expression of TbpB, Slam1, and Skp in the *N. meningitidis* strains examined. Depletion of Skp did not affect the expression of outer membrane protein (OMP) Slam1 or TbpB (induced by 0.1 mM deferoxamine). Fluorescent assay results are combined from three biological replicates and statistically analyzed by one-way analysis of variance (ANOVA) test.

The online version of this article includes the following source data and figure supplement(s) for figure 5:

**Source data 1.** Fluorescent signal of TbpB on the surface of *N. meningitidis.*

**Figure supplement 1.** Deletion of Skp in *N. meningitidis.*

**Figure supplement 2.** Purification of Nme Skp for antibody production and antibody test.

**Figure supplement 3.** Growth of B16B6 *N. meningitidis* strains in BHI media.

**Figure supplement 3—source data 1.** Growth of *N. meningitidis.*

B16B6 *Δskp* strain back to the wild-type level. Taken together, in the absence of the periplasmic chaperone Skp, significantly less properly folded TbpB is translocated to the surface of *N. meningitidis.*

## Discussion

In this study, we illustrate the role of Slam as an outer membrane translocon responsible for the transport of TbpB-like SLPs to the surface of Gram-negative bacteria. By using an in vitro assay to reconstitute the translocation of TbpB across a biological membrane, we showed that Slam1 is necessary to translocate TbpB in absence of other outer membrane machinery such as the Bam complex. Unlike other translocons that require energy such as ATP or proton motive force to mediate translocation, we found that Slam1 instead requires periplasmic chaperones to keep prefolded TbpB available for efficient translocation. Using an *E. coli* model, Skp was found to interact with TbpB and HpuA in the periplasm after these SLPs were released from the inner membrane. Existing in the trimeric form, chaperone Skp is known to act as a holdase for the prefolded OMPs as they localize across the periplasm prior to their insertion into the outer membrane by the Bam complex (*Sklar et al., 2007*; *Mas et al., 2019*). Given that TbpB and HpuA contain at least one beta-barrel domain similar to OMPs (*Calmettes et al., 2012*; *Wong et al., 2015*), Skp might interact with these SLPs in similar manner to keep them in their prefolded states prior to translocation (*Walton et al., 2009*). In our in vitro reconstitution assays, the presence of Skp is important for efficient translocation of TbpB into Slam1 proteoliposomes. The deletion of *skp* in *N. meningitidis* did not affect the expression of the OMP Slam1; however, it resulted in the loss of functional TbpB on the surface of *N. meningitidis*. Although a fraction of TbpB could be detected on the surface in the *Δskp* mutant, it is not clear whether these TbpB were misfolded after the translocation or only a fraction of TbpB was exposed on the surface. Further investigation will be needed to understand how TbpB–Skp complex is recognized and translocated by Slam1, as well as how TbpB is folded once it is localized to the surface. Taken together, our data suggest that periplasmic chaperone Skp is required to keep SLPs in their prefolded states in the periplasm for proper translocation to the surface of Gram-negative bacteria via the Slam translocon.

Combined with our previous work (*Hooda et al., 2016*), we propose a model for the SLPs localization from the inner membrane to the surface of Gram-negative bacteria (*Figure 6*). Upon emerging from the Sec translocon, periplasmic chaperone Skp binds to SLPs and keeps them in the prefolded state. The SLPs are then modified and lipidated by the three enzymes – Lgt, Lsp, and Lnt before being transferred to the Lol complex. As the SLPs are released into the periplasm, LolA accommodates the N-terminal triacyl lipid group (*Tajima et al., 1998*), while Skp remains bound to the SLPs to prevent them from prematurely folding prior to being translocated by Slam. Upon being released to the periplasm, SLP complexes are recognized by Slam for translocation to the cell surface. Drawing from similarities between two-partner secretion (*Guérin et al., 2017*) and the Slam system, we propose the movement of the SLP across the outer membrane occurs via the Slam membrane domain. Interestingly, Slam substrates such as TbpB or HpuA also contain a lipid anchor which most likely needs to be flipped from the inner leaflet of the outer leaflet. The presence of a lateral opening in the Slam transmembrane domain would allow the hydrophobic lipid anchor access to the hydrophobic membrane environment, which is similar to functions of lateral gates observed in BamA (*Noinaj et al., 2013*) and LptD (*Gu et al., 2015*) outermembrane translocons. Given that the Slam translocon does not require ATP as an energy source, we speculate that the folding of SLPs on the surface might provide a

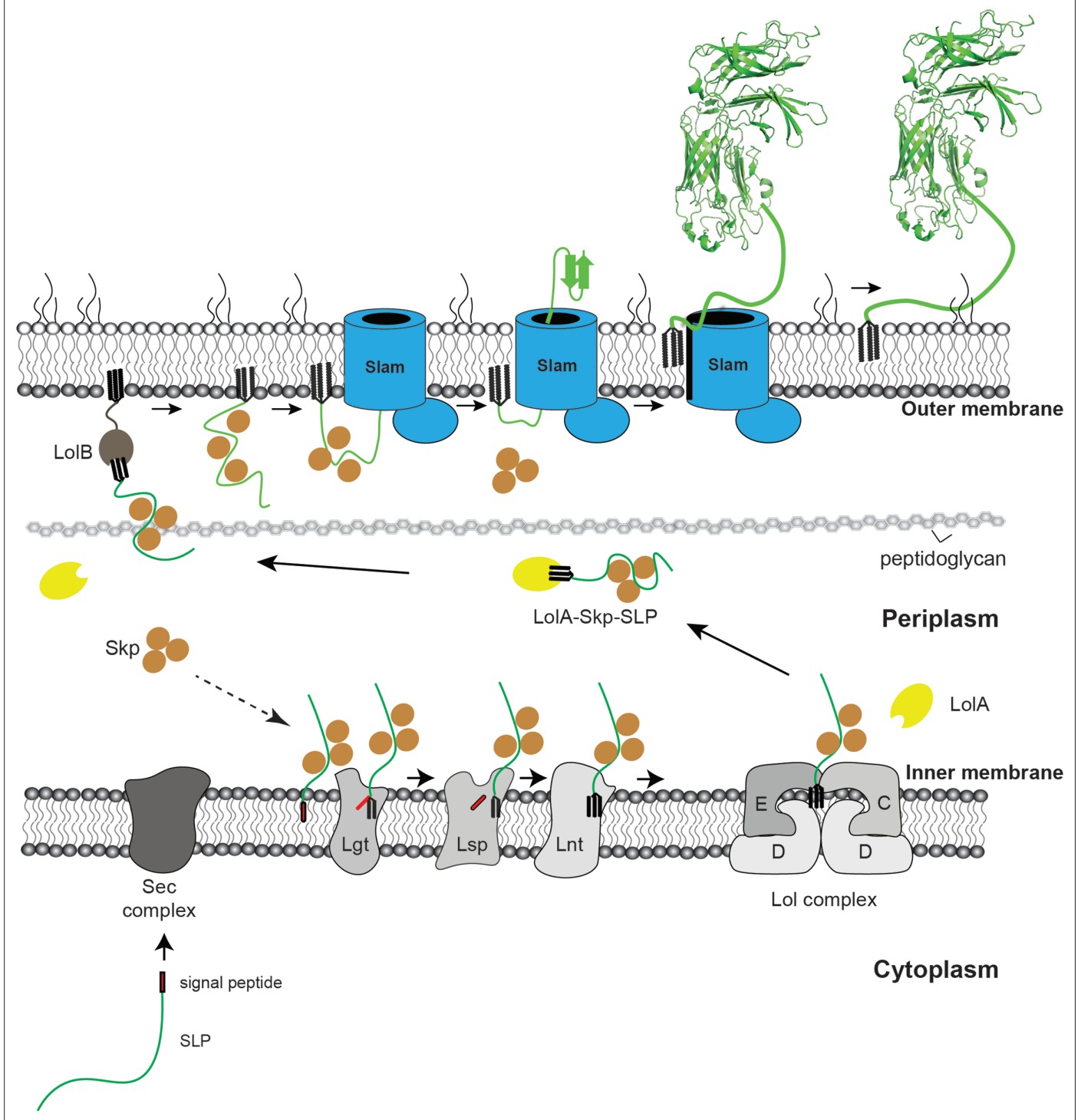

**Figure 6.** Proposed model of localization for Slam-dependent surface lipoproteins in Gram-negative bacteria. Once the surface lipoproteins emerged from the Sec complex, periplasmic chaperone Skp binds to the SLPs to prevent early folding while their N-terminus is modified and lipidated by Lgt, Lsp, and Lnt before getting passed to the Lol complex. LolA then releases lipidated SLPs to the periplasm while Skp stays bound to prevent SLPs folding. LolB in the outer membrane (OM) might serve as the receiver for the LolA–SLP–Skp complex. The 'specificity motif' present at the C-terminus of the unfolded SLP is recognized by Slam and then transported across the outer membrane through the Slam channel and folds on the other side. Once the entire length of the SLP is transported, the Slam lateral gate allows the lipid anchor to 'flip' from the inner leaflet to the outer leaflet of the outer membrane.

driving force to further pull the substrates through Slam barrel domain. High resolution structural and additional biochemical studies of Slam will be required to reveal these mechanistic details of Slam-dependent SLP translocation.

## Materials and methods

### Bacterial strains and growth conditions

Strains used in this study are summarized in *Supplementary file 1*. *E. coli* were grown in LB media containing antibiotics when necessary – 50 µg/ml kanamycin, 50 µg/ml erythromycin, and 100 µg/ml ampicillin. Cloning procedures were carried out using *E. coli* MM294 competent cells. Protein expression was performed using *E. coli* C43 (DE3) cells for Slam homologs, Bam complex, and the translocation experiments (*Wagner et al., 2008*). *E. coli* BL21 (DE3) cells were used for purification of *E. coli* LolA, SurA, Skp, DegP, and B16B6 *N. me* Skp. In vivo translocation reconstitution and sphero-plast secretion assays were performed using *E. coli* C43 (DE3) or *E. coli* K12 cells from Keio's collection (*Baba et al., 2006*). *N. meningitidis* B16B6 strain was used for knockout study.

### Cloning of Slam, SLPs, LolA, and periplasmic chaperones

Genes were cloned into expression vectors by RF cloning (*van den Ent and Löwe, 2006*) and signal peptides and tags were inserted using round the horn cloning (*Liu and Naismith, 2008*). pET52 *Nme hpuA* was made by amplifying *hpua* from *N. meningitidis* strain B16B6 and inserting it into an empty pET52b vector. pET52 *Nme* HpuA-flag was made by addition of a flag-tag at the C-terminus of the *hpua* gene in pET52 *Nme hpuA*. pET26 *Ngo* Slam2 construct was obtained by cloning the mature *N. gonorrhoeae* strain MS11 gene *ngfg_00064* and inserting into an empty pET26b vector. To be expressed in K12 *E. coli*, *slam1*, and *tbpb* were cloned on pGCC4 and pHERD plasmid, respectively. *E. coli lola*, *sura*, *skp*, *degp* genes from *E. coli* strain C43 (DE3) genome and *N. meningitidis skp* gene from *N. meningitidis* B16B6 strain were cloned into an empty pET28a vector with an N-terminal 6xHis tag for purification. The constructs used in this study are summarized in *Supplementary file 1*.

### Plate reader assay for Slam-SLP in vivo translocation assay

Pairs of Slams and SLPs were cotransformed into *E. coli* C43(DE3) or *E. coli* K12 cells. Cells were grown overnight in autoinduction media (*Studier, 2005*) with appropriate antibiotics as described above. Cells were harvested from the overnight culture by centrifugation at 1500 × *g*, 3 min. Cell pellets were washed gently with PBS + 1 mM MgCl$_2$ before incubating with biotinylated human transferrin or rabbit α-flag antibody (1:200 dilution). After 1-hr incubation, cells were harvested and washed with PBS + 1 mM MgCl$_2$. The cells were then incubated with streptavidin-conjugated phycoerythrin (for biotinylated human transferrin) or α-rabbit IgG-conjugated phycoerythrin (for rabbit-α-flag antibody) with 1:200 ratio for 1 hr. Cells were then harvested, washed, and resuspended in PBS + 1 mM MgCl$_2$. The samples were aliquoted on a 96-well plate and read on a Synergy 2 (BioTek) plate reader at 488 and 575 nm. OD600 was also recorded for data normalization.

### Purification of Mcat Slam1

*E. coli* strain C43 (DE3) with pET26 Mcat Slam1 were grown overnight at 37°C in LB + ampicillin. The cells were used to inoculate (1:1000) 6 l of autoinduction media + kanamycin. Cells were grown at 20°C for 48 hr and then harvested by centrifugation at 12,200 × *g* for 20 min at 4°C. The cell pellets were resuspended in 20 ml/l of 50 mM Tris–HCl pH 8, 200 mM NaCl, and cells were lysed using an EmulsiFlex C3 (Avestin). Lysates were spun down at 35,000 × *g* at 4°C for 10 min. The supernatants were spun down in a 45Ti rotor at 40,000 rpm for 1 hr at 4°C to isolate total membranes. Membrane pellets were homogenized, incubated in 15 ml/l of 50 mM Tris pH 8, 200 mM NaCl, 3% Elugent over-night at 4°C and the ultracentrifugation step was repeated to remove insoluble membrane pellet. Supernatants containing the soluble membrane proteins were then incubated with 1 ml Ni-NTA agarose O/N at 4°C. Ni-NTA beads were washed three times with 10 column volumes of buffer A (20 mM Tris pH 8, 100 mM NaCl, 0.03% DDM) containing increasing concentration of imidazole. Mcat Slam1 was then eluted in buffer A containing 200 mM imidazole. The protein sample was exchanged into low salt buffer (20 mM Tris pH 8, 20 mM NaCl, 0.03% DDM) using a PD-10 column (GE Health-care) and then injected onto a MonoQ column (GE Healthcare) equilibrated with low salt buffer. The

column was washed with increasing concentration of salt using a high salt buffer (20 mM Tris pH 8, 2 M NaCl, 0.03% DDM). Fractions that contained pure Mcat Slam1 were identified using SDS–PAGE gels, pooled, concentrated, and stored at −80°C.

## Purification of Bam complex

The plasmid and protocol for Bam complex purification were adapted from Dr. Bernstein's group (*Roman-Hernandez et al., 2014*). *E. coli* strain C43 (DE3) with pJH114 was grown overnight at 37°C in LB + ampicillin. The cells were used to inoculate (1:1000) into 6 l of autoinduction media + ampicillin. Cells were grown at 20°C for 48 hr and harvested by centrifugation at 12,200 × *g* for 20 min at 4°C. Cell pellets were resuspended in 20 ml/l of 50 mM Tris–HCl pH 8, 200 mM NaCl, and cells were lysed using an EmulsiFlex C3 (Avestin). Lysates were spun down at 35,000 × *g* at 4°C for 10 min. The supernatants were spun down in a 45Ti rotor at 40,000 rpm for 1 hr at 4°C to isolate total membranes. Membrane pellets were homogenized, incubated in 15 ml/l of 50 mM Tris pH 8, 200 mM NaCl, 3% Elugent overnight at 4°C, and the ultracentrifugation step was repeated. Supernatants containing the soluble membrane proteins were then incubated with 1 ml Ni-NTA agarose O/N at 4°C. Ni-NTA beads were washed with one column volume with buffer A containing increasing concentration of imidazole. BamABCDE was then eluted in buffer A containing 200 mM imidazole. The protein sample was concentrated and injected onto a S200 column equilibrated with buffer A. Fractions that contained complete BamABCDE complexes were identified using SDS–PAGE gels, pooled, concentrated, and stored at −80°C.

## Liposome and proteoliposome preparation

100 mg of *E. coli* polar lipid extract (Avanti) was resuspended in chloroform (Sigma). The lipid solution was then dried off under $N_2$ gas and resuspended in 10 ml of buffer B (50 mM Tris–HCl pH 7, 200 mM NaCl). The solution was flash frozen and thawed five times and stored at −80°C as a 10 mg/ml stock. For each experiment, 1 ml of the liposome solution (10 mg/ml) was extruded through a 0.2 µm filter (Whatman) to make unilamellar liposomes. The extruded solution was split, and the purified outer-membrane proteins (Bam complex and Slam1 and 2) were diluted 1:5 into the liposome solutions at 1.5 µM for Bam and 15 µM for Slam1 and 2. 50 mg of biobeads SM2 (BioRad) were added to remove detergent and promote protein insertion into liposomes. Tubes were sealed with parafilm and kept at room temperature with gentle end-to-end rotation for ~2 hr. Beads were changed two more times and the proteoliposomes were incubated at 4°C overnight with end-to-end rotation. Proteoliposomes were separated from biobeads and spun down at 18,000 × *g* at 4°C for 5 min. The supernatant was kept at 4°C and used for the experiments within a week. The insertion of Slam1 and 2 and the Bam complex was assessed by SDS–PAGE gels, silver stain, and western blots with α-His antibody.

## Sucrose floatation assay

The protocol used for the sucrose floatation assay was adapted from Dr. Müller's group (*Fan et al., 2012*) with a few modifications. 200 µl of Bam and Slam1 proteoliposomes were resuspended in 1000 µl solution containing 60% sucrose (wt/vol) and transferred to a 5-ml thin-wall polypropylene Beckman tube. The 60% sucrose was layered with 3.8 ml of 30% sucrose and 200 µl of buffer B. The samples were spun in a Beckman SW 50.2 Ti rotor at 45,000 rpm for 16 hr at 4°C. Upon ultracentrifugation, 500 µl fractions were collected from the top. Each fraction was precipitated with 5% TCA, washed three times with 100% acetone. The samples were resuspended in 100 µl of 1× SDS buffer and alternate fractions (first, third, fifth, seventh, and ninth) were run on an SDS–PAGE gel. Western blots were completed with α-His antibody to estimate the quantity of Mcat Slam1 and BamABCDE present in each of the fractions.

## Purification of periplasmic chaperones from *E. coli* and *N. meningitidis*

Purifications were performed similarly for the soluble proteins *E. coli* BL21 (DE3) cells expressing either *E. coli* LolA, SurA, Skp, DegP, or *N. meningitidis* Skp were grown in 20 ml of LB + kanamycin overnight at 37°C and used for inoculating 2 l of 2YT media. The cells were grown at 37°C to an $OD_{600}$ ~ 0.6, induced with 1 mM IPTG and then incubated overnight at 20°C. The cells were harvested the next day by centrifugation at 12,200 × *g* for 20 min at 4°C. The pellets were resuspended in buffer B (50 mM Tris–HCl pH 7, 200 mM NaCl). Cell lysis was performed using EmulsiFlex C3 (Avestin). The

cell lysates were spun down at 35,000 × *g* at 4°C for 50 min to remove cell debris. Supernatant was filtered through a 0.22 µm filter and incubated with 1 ml of Ni-NTA beads for 2 hr at 4°C with gentle stirring. The solution was applied to a column and the Ni-NTA beads were subsequently washed three times with 10 ml buffer B with increasing concentrations of imidazole (10, 20, and 40 mM). Proteins were eluted from the Ni-NTA beads by adding buffer B with 200 mM imidazole. The purified proteins were dialyzed overnight in buffer B at 4°C. The proteins were further purified using S75 or S200 gel filtration (GE Healthcare). The purity of proteins was accessed on SDS–PAGE. The proteins were either stored at −80°C or sent for antibody production.

### Spheroplast release assay

The protocol was adapted from Dr. Müller's group (*Fan et al., 2012*) with a few modifications. Briefly, spheroplasts were obtained from *E. coli* C43 (DE3) or *E. coli* K12 cells transformed with either pET52 Mcat TbpB-flag or pET52 *Nme* HpuA-flag or pHERD Mcat TbpB-flag (*E. coli* K12 only). The cells were grown in LB with 100 µg/ml ampicillin and induced for expression by 0.5 mM IPTG (*E. coli* C43) or 0.1% arabinose (*E. coli* K12) overnight at 20°C. *E. coli* cells were adjusted to have $OD_{600}$ ~ 1.0. The cells were harvested by centrifugation at 6800 × *g* for 2 min at 4°C. The pellets were then resuspended in 100 µl of buffer containing 50 mM Tris–HCl pH 7 and 0.5 M sucrose. The resuspended solutions were kept on ice and converted to spheroplasts by adding 100 µl of buffer containing 0.2 mg/ml lysozyme and 8 mM EDTA with gentle inversion for mixing. The solutions were incubated on ice for at least 20 min. The spheroplasts were collected by spinning at 10,000 × *g* for 10 min and resuspended in 100 µl of M9 minimal media containing M9 minimal salts, 2% glucose and 0.25 µM sucrose. Expression of SLPs was resumed by addition of 0.5 mM IPTG or 0.1% arabinose. 10 µM of *E. coli* LolA was added to promote the release of SLPs from the spheroplasts at 37°C. Samples were collected at different time points and spun down at 18,000 × *g* for 10 min at 4°C to remove spheroplasts. Supernatants at different time points were mixed with SDS loading buffer and run on an SDS–PAGE gel. Western blot analysis using α-flag antibody to estimate the quantity of TbpB and HpuA released by spheroplasts upon the addition of LolA.

### Bam complex functional assay

To test the activity of the Bam complex, the ability of Bam proteoliposomes to potentiate the insertion of spheroplast-released OmpA was used. *E. coli* strain C43 (DE3) cells were converted into spheroplasts and recovered in M9 minimal media as previously described. Spheroplasts were then spun down at 18,000 × *g* at 4°C for 10 min to isolate the secreted supernatant. Supernatant was spun down again at 60,000 × *g* at 4°C to further remove insoluble and remains of outer membrane. Top 200 µl of the soluble fraction was collected and kept on ice. 10 µl of iced-cold supernatant was incubated with either 10 µl of buffer B (liposome buffer), empty liposome, or Bam proteoliposome. Incubations were started every 5 min and all reactions were stopped at the same time by adding 5 µl of 5× SDS loading buffer. Samples of 0, 5, 10, and 20 min were loaded on SDS–PAGE and followed by α-OmpA western blot to access the folding process of *E. coli* OmpA in the presence of Bam proteoliposome.

### Purification of Mcat TbpB

*E. coli* C43 (DE3) cells were transformed with pET52b Mcat TbpB flag-tag. The cells were grown in 20 ml of LB + 100 µg/ml ampicillin overnight at 37°C and were used to inoculate 2 l of 2YT + 100 µg/ml ampicillin the next day. Once $OD_{600}$ reached 0.6, 1 mM IPTG was added to induce Mcat TbpB-flag and the protein expression was carried overnight at 20°C. The purification was performed similar to Slam and Bam OMP purification protocol. After the membranes were extracted and solubilized in 50 mM Tris pH 8, 200 mM NaCl, and 0.1% DDM, 100 µl of flag-beads (sigma) was added into the solution and incubated for 4 hr at 4°C. The beads were loaded on a gravity column and washed three times with 5 ml of 50 mM Tris pH 8, 200 mM NaCl, and 0.03% DDM. Mcat TbpB-flag was eluted by adding 500 µl of 0.1 M glycine, pH 3.5, 0.03% DDM, and 100 µl of 1 M Tris pH 8 was immediately added into the eluted fraction. A280 of the last eluted droplet was measured to determine whether additional volume is needed to elute more protein. All eluted fractions were pooled and concentrated to 0.5 mg/ml. The protein was flash-frozen in liquid nitrogen and stored at −80°C for in vitro proteoliposomes assay.

### Dot blot assay for testing function of TbpB

0.5 µl of TbpB (1 mg/ml), TbpA (1 mg/ml), BSA (1 mg/ml), and BamABCDE (1 mg/ml) was spotted on a nitrocellulose membrane. The cells were blocked with 5% skim milk and then developed with a biotinylated human transferrin (50 µg/ml) followed by streptavidin-conjugated HRP.

### Translocation assay with purified TbpB

To develop the defined translocation assay, purified TbpB was diluted to 6 µM in buffer B or 8 M urea. The TbpB samples were rapidly diluted 1/12 into 50 µl of Empty, Bam, Slam1&2, and Bam + Slam1&2 proteoliposomes to bring the final concentration of TbpB to 0.5 µM and urea to 0.66 M. The samples were incubated for 15 min at 37°C with addition of 10 mg biobeads. The solutions were isolated and then incubated with proteinase K (0.5 mg/ml) in the presence or absence of Triton X-100 (1%). Samples were incubated at room temperature for 30 min. 5 mM PMSF was then added to inhibit proteinase K. Samples were then run on SDS–PAGE, followed by western blotting and α-flag antibody was used to detect TbpB.

### Spheroplast-dependent translocation assay

To develop the spheroplast-dependent translocation assay, we followed the protocol described above for the generation of spheroplasts. Spheroplasts were collected by spinning at 10,000 × $g$ for 10 min and resuspended in 100 µl of M9 minimal salt media containing M9 minimal salts, 2% glucose, 0.25 µM sucrose, and 10 µM of *E. coli* LolA. Subsequently, 50 µl of empty liposomes or Bam, Slam1, or Bam + Slam1 proteoliposomes were added to the separate tubes of the spheroplasts. Expression of TbpB was induced by the addition of 1 mM IPTG and incubation at 37°C for 15 min. Spheroplasts were spun down at 18,000 × $g$ for 10 min at 4°C. Supernatants were collected and treated with the final concentration of 0.5 mg/ml proteinase K in the presence/absence of 1% Triton X-100 and incubated at 37°C for 1 hr. 5 mM PMSF was added to inactivate the proteinase K and samples were loaded on SDS–PAGE gels followed by western blots with α-flag antibodies to assess protection from proteinase K activity.

### Spheroplast-independent translocation assay

A similar protocol was performed for spheroplast-independent translocation assay. After 30 min of spheroplasts resuming protein expression in M9 media with addition LolA, the solution was spun down at 16,000 × $g$ for 10 min at 4°C. 50 µl of obtained supernatant was incubated with 50 µl of Empty, Bam, Slam1, or Bam + Slam1 proteoliposomes for additional 15 min at 37°C (1:1). The samples were then treated with proteinase K (0.5 mg/ml) in the presence or absence of Triton X-100 (1%) as described in the previous section.

### TbpB pulldown assay

C-terminal flag-tagged TbpB was released from *E. coli* spheroplasts as described above. After 15 min of incubating with LolA, spheroplasts were removed by spinning down at 16,000 × $g$ for 20 min at 4°C. 1 ml of supernatant was obtained and incubated with 50 µl prewashed flag beads at 4°C for 2 hr. Beads were spun down at 700 × $g$ at 4°C for 10 min and supernatant was collected as flow through (FT). Beads were washed three times with 1 ml of 1× M9 media. Bead samples were sent to mass spectrometry facility (SPARC – Sickkids) for trypsin digestion and analysis. Data analysis was done by Scaffold 4 software. Cytoplasmic contaminations and proteins that have less than five total spectrums count were excluded from the summary table. For eluting protein complexes, beads were incubated with 200 µl of 50 mM glycine pH 2.8 at room temperature for 5 min. Beads were spun down at 700 × $g$ at 4°C for 10 min and supernatant was collected as elution (E). All samples were treated with 5× SDS loading buffer and pH was adjusted before loading on SDS–PAGE followed by western blotting. TbpB and AfuA (the negative control) were detected using rabbit α-flag antibody, followed by α-rabbit HRP secondary antibody. LolA was detected using mouse α-his antibody and Skp was detected using mouse α-*E. coli* Skp antibody, followed by α-mouse HRP secondary antibody.

### Chaperone pulldown assays

His-tagged chaperones (SurA, Skp, and DegP) were purified as described above. 10 µM of each chaperone was added along with 10 µM untagged LolA during TbpB/HpuA expression in *E. coli*

spheroplasts. After 15 min, spheroplasts were removed by spinning down at 16,000 × $g$ for 20 min at 4°C. 1 ml of supernatant was obtained and incubated with 20 µl prewashed Ni-resin at 4°C for 2 hr. Beads were spun down at 700 × $g$ at 4°C for 10 min and supernatant was collected as flow through (FT). Beads were washed three times with 1 ml of 50 mM Tris 7, 200 mM NaCl, 10 mM imidazole, and 0.1% Triton X-100. Proteins were eluted with 100 µl of 50 mM Tris 7, 200 mM NaCl, and 200 mM imidazole. All samples were treated with 5× SDS loading buffer and pH was adjusted before loading on SDS–PAGE followed by western blotting. TbpB, HpuA, and AfuA (the negative control) were detected using rabbit α-flag antibody, followed by α-rabbit HRP secondary antibody.

## Reconstitution of Mcat Slam1 and Mcat TbpB in K12 *E. coli* strains (wild type and mutants)

*E. coli* K12 wild type, K12 Δ*skp* and K12 Δ*degp* were obtained from the Keio's collection (*Baba et al., 2006*). These cells were cotransformed with pGCC4 *mcat slam1* (with N-terminal his-tag) and pHERD *mcat tbpb* (with C-terminal flag-tag). Successfully transformed cells were selected on LB + erythromycin (50 µg/ml) + ampicillin (100 µg/ml) plate. Cells were grown in LB media with the appropriate antibiotics until $OD_{600}$ ~ 0.6 and then were treated with 0.5 mM IPTG for Slam1 overnight expression. The next day, the cells were spun down at 3000 rpm for 5 min and the pellets were resuspended in fresh LB media (with appropriate antibiotics), recovered for 30 min at 37°C, 150 rpm. 0.1% arabinose was added into the media to induce the expression for TbpB for 4 hr. The cells were then harvested and ready for plate reader assay with biotinylated human transferrin and α-flag antibody as previous described above.

## In vitro proteoliposomes translocation with addition of periplasmic chaperones

The assay was modified based on the previous assay described above for purified TbpB. In this assay, 10 µM of DDM-Mcat TbpB complex was diluted 1:10 in 50 mM Tris 7, 200 mM NaCl, 8 M urea buffer with addition of 20 mg biobeads, 10 µM *E. coli* LolA and 30 µM *E. coli* Skp or *E. coli* SurA (negative control). The denaturation was performed at 4°C for 2 hr in 1.5 ml microcentrifuge tube with end-to-end rotation. The beads and insoluble were removed by spinning down at 16,000 × $g$ for 5 min. 50 µl of the supernatant was then incubated with 250 µl of either empty liposomes or Slam1 proteoliposomes (to further dilute urea concentration) with 50 mg fresh SM2 biobeads. The solutions were incubated at room temperature for 1 hr and were then treated with 0.1 mg/ml proteinase K or proteinase K + 0.1% Triton X-100 for 15 min. 1 mM PMSF was added to inhibit the proteinase K before adding SDS loading buffer for gel electrophoresis and western blot.

For the follow-up sucrose floatation assay, the proteinase K digested proteoliposomes solutions (no Triton X-100 treatment) were mixed with 1 ml of 60% sucrose and incubated on ice for 10 min. A layer of 10 ml of 30% sucrose was then added on top and incubated on ice for 10 min. 1 ml of 50 mM Tris 7, 200 mM NaCl was used to top up the 13 ml polyethylene tube and the solutions were spun at 27,000 × $g$ for 18 hr at 4°C using SW45 rotor (Beckman). 1 ml of top fraction, 10 ml of middle fraction, and 2 ml of bottom fractions were collected for TCA precipitation (*Koontz, 2014*). The pellets were resuspended in 1× SDS loading buffer, followed by SDS–PAGE and α-flag western blots to estimate the quantity of Mcat TbpB in each fraction.

## Gene deletion and complementation of Skp in *N. meningitidis*

Restriction-free (RF) cloning was used for the following plasmid (*Supplementary file 1*). To completely replace *skp* gene with a kanamycin cassette, pUC19 Δ*skp*::kan plasmid was cloned to contain the *kan2* gene with upstream and downstream 500 bp flanking region of *skp*. The plasmid was used to transform *N. meningitidis* B16B6 strain using spot transformation on BHI plate (*Dillard, 2011*). The plate was incubated overnight at 37°C, 5% $CO_2$. The lawn within the spot was streaked onto a BHI + 75 µg/ml kanamycin and incubated for 18 hr. Colony PCR was used to select cells that have *skp* deleted and the colony was then grown in 3 ml of BHI media + 75 µg/ml kanamycin overnight at 37°C, 5% $CO_2$. The cells were adjusted to have $OD_{600}$ ~ 1.0 and 500 µl was spun down at 3000 rpm for 5 min while the remaining cells were used to make 30% glycerol stock and stored at −80°C. The cell pellets were resuspended in PBS buffer. 2× SDS loading buffer was then added for SDS–PAGE, followed by an α-*Nme* Skp antibody to confirm the absence of Skp in the B16B6 Δ*skp* mutant.

Complementation vector pGCC4 Nme Skp was constructed by cloning the B16B6 *skp* gene into the PacI/FseI site of pGCC4 by RF cloning. The plasmid was used to transform B16B6 *N. meningitidis* Δ*skp* strain using spot transformation. The lawn within the spawn was streaked onto a BHI + 5 µg/ml erythromycin plate and incubated for 36 hr. Colony PCR was used to select cell colonies that have *skp* gene reintroduced. The colonies were then streaked on new BHI (+5 µg/ml erythromycin) with 1 mM IPTG plate and incubated overnight. Colonies were collected, resuspended in 1× SDS loading buffer, ran on SDS–PAGE, and transferred on PVDF blots. α-Nme skp antibody was used to access the expression of Skp from pGCC4 plasmid in the B16B6 Δ*skp* mutant.

### *N. meningitidis* growth assay

B16B6 *N. meningitidis* wild type, Δ*slam1*, Δ*tbpb*, and Δ*skp* mutant was grown overnight in 2 ml BHI ± 50 µg/ml kanamycin. The $OD_{600}$ was adjusted to 1.0 and 2 µl was used to inoculate 200 µl of BHI ± 50 µg/ml kanamycin. The cells were grown on a 96-well plate with 150 rpm shaking at 37°C. The $OD_{600}$ was recorded every 30 min for 24 hr using Nivo microplate reader (VICTOR Nivo).

### Exposure of functional TbpB on the surface of *N. meningitidis* mutants

The cultures were started similar to the growth assay. After adjusting the $OD_{600}$ to 1.0, 30 µl of cells were used to inoculate 3 ml of BHI ± kanamycin (50 µg/ml) in a 15-ml culture tube. After 4 hr, 0.1 mM deferoxamine was added to induce expression of TbpB. 1 mM IPTG was also added to Δskp + pGCC4 Nme Skp to induce expression of Skp. The cells were grown for 16 hr at 37°C, 5% $CO_2$. Cells were adjusted to have $OD_{600}$ ~ 1.0 and 1 ml of cells were spun down at 3000 rpm for 5 min. Cell pellets were washed with 500 µl of PBS + 1 mM $MgCl_2$ and then resuspended in 200 µl of PBS + 1 mM $MgCl_2$ + 50 µg/ml biotinylated human transferrin (bio-htf) or rabbit α-TbpB antibody (1:200 of unknown concentration) followed by 1-hr incubation at 25°C. The cells were spun down at 3000 rpm for 5 min and the pellets were washed three times with 200 µl of PBS + 1 mM $MgCl_2$. The cell pellets were resuspended in 200 µl of PBS + 1 mM $MgCl_2$ buffer with 50 µg/ml streptavidin-conjugated phycoerythrin (for primary of bio-htf) or 50 µg/ml α-rabbit IgG-linked phycoerythrin (for primary of α-TbpB) and incubated for 1 hr. Cells were spun down, pellets were washed three times, resuspended in 200 µl of PBS + 1 mM $MgCl_2$ and transferred into Greiner 96-well plates black flat-bottom. Fluorescence intensity was read using a microplate reader (Synergy) at wavelength 488 nm (excitation) and 575 nm (emission). $OD_{600}$ was measured for normalizing the fluorescent signal.

### Materials and correspondence

All data are available in the main text or the supplementary materials. Correspondence and requests for materials should be addressed to trevor.moraes@utoronto.ca.

## Acknowledgements

OmpA antibodies were obtained from Dr. Jan Willem deGier, Stockholm University. Plasmid for expression of the Bam complex (BamA–E) was obtained from Dr. Harris Bernstein at the NIH. Mr. Ashutosh Gupta, Andrew Judd, and River Jiang helped in development of the purification protocol for Slams. Funding for this study was obtained from the Canadian Institutes of Health Research (CIHR PJT-148795).

## Additional information

### Competing interests

Yogesh Hooda, Christine Chieh-Lin Lai, Trevor F Moraes: is a co-author on a patent, 'Slam polynucleotides and polypeptides and uses thereof' - patent number WO2017136947A1. The other authors declare that no competing interests exist.

## Funding

| Funder | Grant reference number | Author |
|---|---|---|
| Canadian Institutes of Health Research | PJT-148795 | Trevor F Moraes |

The funders had no role in study design, data collection, and interpretation, or the decision to submit the work for publication.

## Author contributions

Minh Sang Huynh, Conceptualization, Data curation, Formal analysis, Methodology, Writing – original draft, Writing – review and editing; Yogesh Hooda, Conceptualization, Data curation, Methodology, Writing – original draft, Writing – review and editing; Yuzi Raina Li, Data curation, Methodology; Maciej Jagielnicki, Methodology, Writing – review and editing; Christine Chieh-Lin Lai, Data curation, Methodology, Writing – review and editing; Trevor F Moraes, Conceptualization, Formal analysis, Funding acquisition, Supervision, Writing – review and editing

## Author ORCIDs

Minh Sang Huynh http://orcid.org/0000-0002-9541-6441
Trevor F Moraes http://orcid.org/0000-0001-9883-6145

## Decision letter and Author response

Decision letter https://doi.org/10.7554/eLife.72822.sa1
Author response https://doi.org/10.7554/eLife.72822.sa2

---

# Additional files

## Supplementary files

• Supplementary file 1. Strains, plasmids, and antibodies used in this study. Raw images for figures and figure supplements.

• Transparent reporting form

• Source data 1. Mass spectrometry results.

• Source data 2. Original gels and blots used for the figures.

## Data availability

All data generated or analyzed during this study are included in the manuscript and supporting files.

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
