## [Editor Report]

This work elucidates the pathway of how surface-exposed lipoproteins of Gram-negative bacteria reach their destination in the outer membrane. Authors have identified an outer membrane protein complex that serves as a translocon for the lipoproteins and discovered the pivotal role of a periplasmic chaperone in the targeting pathway. This work will provide new insights into host invasion mechanisms by pathogenic bacteria, in which surface lipoproteins are critically involved.

---

## [Decision Letter]

**Decision letter after peer review:**

Thank you for submitting your article "Reconstitution of surface lipoprotein translocation reveals Slam as an outer membrane translocon in Gram-negative bacteria" for consideration by *eLife*. Your article has been reviewed by 3 peer reviewers, one of whom is a member of our Board of Reviewing Editors, and the evaluation has been overseen by Gisela Storz as the Senior Editor. The reviewers have opted to remain anonymous.

Overall, reviewers agree that the reconstitution of Slam-mediated protein translocation potentially has a high impact and the identification of Skp as a specific vehicle for proper targeting and assembly is a novel finding.

We hope that authors will constructively address the points raised by the reviewers.

Essential revisions:

I. Major scientific concerns:

1) Please further emphasize the novelty of this work in a more persuasive way for general readers and researchers in the field.

Reviewer 1 raised critical concerns on the rationale of this work. More specifically,

1-i) Have we learned novel molecular insights besides the confirmation of the group's previous discovery?

1-ii) The specific role of Skp in translocation is interesting but not surprising, considering that periplasmic holdases are already known to be extensively involved in the biogenesis of periplasmic and outer membrane proteins.

2) To claim that TbpB has been indeed translocated by Slam in the liposomal system, more convincing data need to be provided in addition to Proteinase K (PK) protection. Reviewers ask the following specific questions:

2-i) Does the portion of TbpB that is protected from PK have correct conformation and membrane-topology? (Reviewers 1 and 3)

2-ii) How can authors rule out the possibility that binding of TpbB to Slam1 does not protect TbpB from PK digestion? (Reviewer 3)

2-iii) Showing the time-dependence of the translocation process and comparing it to the timescale of other reported translocation processes across the outer membrane (for example, membrane-insertion rates of BAM-mediated OMP translocation in vitro) will further support authors' conclusion that TbpB is translocated across the outer membrane. (Reviewer 3)

2-iv) Please consider using a model unfolded protein other than TpbB as a control in the translocation assay to verify the specificity of Slam for TpbB. (Reviewer 3)

3) Please, address the following questions that can be potentially raised by readers.

3i) Is the lipidation of TpbB required for translocation? (Reviewer 3)

3ii) Figure S2: a permeabilization control should be included: what if Slam1 overexpression changes the permeability of the OM? (Reviewer 3)

4) Please, move the data demonstrating the functionality of BamA using OmpA as a model substrate (Supplementary Figure 4) to the main figures. (Reviewer 2)

*Reviewer #1 (Recommendations for the authors):*

1. Lines 119-127: More explanations are needed regarding what are the effects of urea on liposomes, conformation of TbpB, Slam1 and Bam:

>Is 0.66M urea enough to keep TbpB unfolded and Slam1 and Bam still functional?

>Would there be any osmotic stress on liposomes, which may affect the translocation of TbpB?

>Is 3% translocation efficiency enough to say that the translocation has actually occurred?

2. Figure 1b: More explanations are needed on the band patterns:

> Why is the TbpB band split into three-two at ~75 kD and one at ~63 kD (+ProK and -Triton) in the presence of urea?

> The additional weak lower band of TbpB (~63 kD, +Proteinase K and -Triton) should be explained as in the asterisk-marked Figure 2b.

3. Line 125:

I am not sure whether the usage of the term "pore" is warranted considering there is no structural evidence on the existence of the pore yet.

4. Figure 2a and Lines 132-134: Lgt, Lsp and Lnt as well as the associated processes should be explained.

5. Line 203: "speroplast" is a typo; Line 213: "wildtype" should be changed to wild type (there are the same typo in many places)

6. Lines 206-207: "Depletion of DegP slightly reduced the translocation of TbpB but this was not statistically significant."

> As mentioned in the point (4a) in Public Review, Figure 3 and 4 indicate some contribution of DegP on the productive translocation of TbpB or HpuA (although weak). More explanation is needed to completely exclude the contribution of DegP.

7. Lines 248-251: "The western blots and Coomassie blue stained SDS-PAGE showed a clear separation of the two components (Figure 4d). While most of the proteins were in the bottom and middle fraction, significant amount of Slam1 and TbpB were found only in the liposomes fraction collected from the top layer."

> Figure 4d is not fully convincing. If there is no Slam1 in liposomes (empty), Tbp or Tbp/Skp proteins should be separated from the liposomes and stay at the bottom. But, there is no Tbp detected at the bottom.

8. Line 277: An explanation is needed for the B16B6 Δskp strain. It popped up out of nowhere.

9. Skp is known as a highly basic protein and electrostatic interaction are known to be important for binding of clients. It should be discussed whether the sequence of TbpB has any local or global sequence features for interacting with Skp.

10. Lines 322-323: "Upon their insertion into the inner leaflet of the outer membrane, LolA:SLP:Skp complexes are recognized by Slam for translocation to the cell surface."

> There seems no evidence regarding whether the LolA:SLP:Skp complexes bind the outer membrane first and then are recognized by Slam. They could be directly recruited by Slam.

11. Lines 330-332: "Given that Slam translocon seems to require no energy input, we speculate that the folding of SLPs on the surface might provide driving force to further pull the SLPs through Slam barrel domain."

> This claim (no energy input is required) may be toned down because the reconstitution system always yields low translocation efficiency compare to the cell-based system.

12. Misleading figure calling

12a) Lines 195-199: "To further validate the interaction between Skp and SLPs, a reciprocal pulldown assay was performed in which a purified His-tagged chaperone was added into the spheroplast prior to the secretion of SLPs. In this assay, we also examined whether Skp interacts

with other SLPs such as hemoglobin-haptoglobin utilization protein (HpuA) – a substrate of Slam2 homolog in N. meningitidis (Hooda et al., 2016) (Figure 3b)."

> This sentence should designate Figure 3c/d not Figure 3b.

12b) Lines 210-213: "The presence of TbpB on the surface of *E. coli* was detected using rabbit α-flag antibody, followed by phycoerythrin-conjugated α-rabbit IgG which fluoresces at 575nm. The results showed that only *E. coli* K12 Δskp mutant had significant reduction of TbpB's surface exposure (50%) compared to wildtype cells."

> Figure calling should be done after this sentence.

12c) Line 253: "this ratio is consistent with the previous result if accounting for the 2-fold protection coming from the Skp-TbpB interaction."

> "The previous result" should be specified (Figure number? Reference?)

13. Missing Figure numbering: "Figure 2"

> There are legends for Figure 2c but the actual Figure 2c does not exist in the figure panel.

*Reviewer #2 (Recommendations for the authors):*

Please see my comments in the Public Review about demonstrating that the Bam complex they have purified and include in proteoliposomes is functional.

The legends and labelling of figures should be revised to clarify assays and results.

For instance in Figure 1 SM2 beads/DDM are not explained in the results or the legend, the numbers under the western blots are not explained, and the Y axis in panel C gives % TbpB insertion while the legend describes protection.

*Reviewer #3 (Recommendations for the authors):*

Although I agree that the results presented in the current manuscript support the idea that Slam1 mediates the translocation of TpbP, I think that the authors do not have enough data to make that claim. For instance, how can they rule out that binding of TpbB to Slam1 does not protect TbpB from PK digestion?

They should 1) show the time-dependence of the translocation process; 2) identify a Slam1 mutant that does not function in vivo and show that when used in the in vitro assay, it does not function either; 3) obtain evidence that TpbP is folded once inside the proteoliposomes.

Thus, additional data is required to make a fully convincing case.

The following points should also be addressed:

-an unfolded protein other than TpbB should be used as a control in the translocation assay

-determine whether lipidation of TpbB is required for translocation

-Sup. Figure 2: a permeabilization control should be included: what if Slam1 overexpression changes the permeability of the OM

---

## [Author Response]

Essential revisions:I. Major scientific concerns:1) Please further emphasize the novelty of this work in a more persuasive way for general readers and researchers in the field.Reviewer 1 raised critical concerns on the rationale of this work. More specifically,1-i) Have we learned novel molecular insights besides the confirmation of the group's previous discovery?1-ii) The specific role of Skp in translocation is interesting but not surprising, considering that periplasmic holdases are already known to be extensively involved in the biogenesis of periplasmic and outer membrane proteins.

The role of Slam in the translocation of lipoproteins to the surface of bacterial cells was not addressed in the Hooda et al. discovery of Slam. Here we demonstrate that Slam is autonomously able to translocate proteins across membranes. To date there has been no study has demonstrated the interaction of Skp with lipoproteins prior to translocation across the outer membrane in a Slam-dependent manner. So far, there are only studies describing Skp as a holdase for OMPs (transmembrane only) and quality control for unfolded proteins in the periplasm. Furthermore, LolA is the only known periplasmic factor that binds to the triacyl-lipid at the N-terminal of lipoproteins in these biogenesis studies. The fate of lipoproteins in the periplasm in term of stability, folding or interactions have not been studied. What we propose in this paper is that LolA binds to the triacyl-lipid and the chaperone Skp binds the surface lipoproteins in the periplasm to prevent them from early folding for more efficient and functional translocation via the Slam outer membrane protein.

2) To claim that TbpB has been indeed translocated by Slam in the liposomal system, more convincing data need to be provided in addition to Proteinase K (PK) protection. Reviewers ask the following specific questions:2-i) Does the portion of TbpB that is protected from PK have correct conformation and membrane-topology? (Reviewers 1 and 3)

TbpB is not transmembrane protein, therefore we did not look into TbpB membrane topology after translocation. However, we have performed a functional pulldown assay for the translocated TbpB in the Slam-proteoliposomes using human transferrin conjugated beads to show that these TbpB molecules are correctly folded and functional. Blots and explanation are attached in the revised manuscript (see Figure 2 —figure supplement 2 and discussion in main text line 197-207).

2-ii) How can authors rule out the possibility that binding of TpbB to Slam1 does not protect TbpB from PK digestion? (Reviewer 3)

To address this question we have performed Proteinase K digestion using purified Slam, TbpB in DDM detergent (Slam-DDM-TbpB). We have included a new supplementary (Figure 1—figure supplement 7 – upper panel and line 133-136). The Slam-DDM-TbpB protein mixture did not protect TbpB from the proteinase K digestion. Also, in our liposome translocation assay, TritonX was used as a control to dissolve liposomes and reveal the contents. In the PK^+^/T+ treatment, the portion of TbpB that was seen protected from PK treatment (no TritonX) disappeared indicating that the liposomes protected TbpB rather than Slam (as TritonX dissolves liposomes).

2-iii) Showing the time-dependence of the translocation process and comparing it to the timescale of other reported translocation processes across the outer membrane (for example, membrane-insertion rates of BAM-mediated OMP translocation in vitro) will further support authors' conclusion that TbpB is translocated across the outer membrane. (Reviewer 3)

We have performed the time-dependent translocation assay as recommended by the reviewers and the included the result in updated as Figure 2d and 2e, line 211-224. The time course reveals a peak translocation time of 15-30min on a similar timescale to the Bam complex for membrane protein insertion.

2-iv) Please consider using a model unfolded protein other than TpbB as a control in the translocation assay to verify the specificity of Slam for TpbB. (Reviewer 3)

We have performed a negative control translocation assay using the periplasmic binding protein AfuA – a glucose-6-phosphate binding protein that is localized to the periplasm. The blot is included in Figure 1 —figure supplement 7 – lower panel and line 136-140.

3) Please, address the following questions that can be potentially raised by readers.3i) Is the lipidation of TpbB required for translocation? (Reviewer 3)

The lipidation of TbpB is not required for translocation as i) We have other Slam-secreted proteins (non-lipidated proteins) pairs that our lab is currently studying (See Bateman et al. Nature Communications 2021). ii) We have data to support that removing the N-terminal cystine residue (where the lipid is added on by Lgt) does not affect the translocation of TbpB by Slam and the non-lipidated TbpB can be now secreted into the extracellular side rather that getting anchored to the surface (this data will be presented in an upcoming paper by another member of the lab).

3ii) Figure S2: a permeabilization control should be included: what if Slam1 overexpression changes the permeability of the OM? (Reviewer 3)

In the Figure S2, we also have included Slam2-TbpB pair as a control where the expression of a Slam homolog does not increase lead to TbpB exposure due to permeability (This figure is now Figure 1 —figure supplement 2). This was also addressed in Hooda et al. Nature Microbiology 2016 where the Slam2 homolog of Slam1 does not translocate TbpB. If the overexpression of Slam affects the permeability of the outer membrane, we should also expect a high portion of TbpB displayed by Slam2 too in this assay.

4) Please, move the data demonstrating the functionality of BamA using OmpA as a model substrate (Supplementary Figure 4) to the main figures. (Reviewer 2)

We believe that the functionality of Bam has been established in previous papers and prefer to leave this control (Bam functional assay) in the Figure 1 —figure supplement 4 (updated according to e*Life* format) as we want to focus the main figures on Slam.

Reviewer #1 (Recommendations for the authors):1. Lines 119-127: More explanations are needed regarding what are the effects of urea on liposomes, conformation of TbpB, Slam1 and Bam:>Is 0.66M urea enough to keep TbpB unfolded and Slam1 and Bam still functional?

The protocol used herein was adapted from a similar protocol used to examine urea-OmpA incorporation into Bam proteoliposomes (Hagan et al. 2010) and Bam has been shown to be functional at 0.66M urea. We have used a Nanotemper Tycho instrument to measure the unfolding event of protein (TbpB) in the presence of 8M, 4M, 1M and 0.66M urea illustrating that TbpB is unfolded in 8M Urea and begins to fold at 0.8 M urea (for reference information on the Tycho instrument see: https://resources.nanotempertech.com/tycho/nanotemper-tycho-brochure).

The melting temperatures (indicative of folding) illustrate that TbpB is at least partially folded at 0.66M Urea. We recognize that at this concentration of Urea the interactions between TbpB and Slam may not be optimal and may not approach the ~40% translocation efficiency from supernatant extracts (figure 2) but we can still increase translocation in the presence of skp showing that holdases due also influence the translocation efficiency.

>Would there be any osmotic stress on liposomes, which may affect the translocation of TbpB?

We don’t believe there would be any osmotic stress imposed in the in vitro translocation assay as the same buffer that was used to make the liposomes was also used during the in vitro translocation creating an isotonic solution. Thus, there would be no electrochemical gradient across the lipid bilayer, except for the TbpB concentration that would be higher on the outside of the liposomes.

For the spheroplast release assay, the media might have additional factors including the 2% of glucose and 0.125M sucrose in the extractions, but if they are the major contribute factors, we would have seen similar contribution in the other 4 experimental controls (empty, Bam, Slam1 and Slam1+Bam proteoliposomes).

>Is 3% translocation efficiency enough to say that the translocation has actually occurred?

Yes, when we compare the 3% of the unfolded TbpB translocated by Slam to the negative control (empty and Bam proteoliposomes), our analysis suggested that this 3% is statistically significant. We have included the more detailed analysis in Figure 1c and line 130-133.

2. Figure 1b: More explanations are needed on the band patterns:> Why is the TbpB band split into three-two at ~75 kD and one at ~63 kD (+ProK and -Triton) in the presence of urea?> The additional weak lower band of TbpB (~63 kD, +Proteinase K and -Triton) should be explained as in the asterisk-marked Figure 2b.

In this assay, TbpB has a flag-tag on its C-terminus and we have previous data (Hooda et al. 2016 Nat Micro) and new preliminary data showing Slam recognizes the C-terminus of TbpB for translocation (the new data was not included in this paper as it is under preparation for an upcoming paper on Slam-SLP specificity). We suspect that some translocation was still occurring while being treated with proteinase K and thus these TbpB was degraded before being fully translocated by Slam -this is now indicated with an * a band at ~63 kD (+ProK and -Triton) in the presence of urea.

The three bands of TbpB -(two at ~75 kD and one at ~63 kD in the input) represent degradation from the labile N-terminus of TbpB during purification.

We have briefly explained this in the figure legend of figure 1b.

3. Line 125:I am not sure whether the usage of the term "pore" is warranted considering there is no structural evidence on the existence of the pore yet.

Thank you for your recommendation. We have rewritten this sentence and avoid to use the term “pore” – line 143.

4. Figure 2a and Lines 132-134: Lgt, Lsp and Lnt as well as the associated processes should be explained.

We were afraid of being repetitive, as some of this information (the roles of Lgt, Lsp and Lnt) have been explained in the introduction, but we have rewritten this as recommended (line 152-155 and figure legend of Figure 2a).

5. Line 203: "speroplast" is a typo; Line 213: "wildtype" should be changed to wild type (there are the same typo in many places)

Thank you for catching that, we have fixed Wildtype to wild type (now line 264) and all other wildtype to wild type and the typos “speroplast” to spheroplast (now line 252).

6. Lines 206-207: "Depletion of DegP slightly reduced the translocation of TbpB but this was not statistically significant."> As mentioned in the point (4a) in Public Review, Figure 3 and 4 indicate some contribution of DegP on the productive translocation of TbpB or HpuA (although weak). More explanation is needed to completely exclude the contribution of DegP.

We excluded DegP as a contributor since the pulldown assays showed no interaction with TbpB when the DegP active site is mutated. Also, the Prism one-way ANOVA test performed show no statistical significance in the slightly reduced translocation. As explained above, we can include the analysis report if it is requested. We have added more explanation in line 264-266.

7. Lines 248-251: "The western blots and Coomassie blue stained SDS-PAGE showed a clear separation of the two components (Figure 4d). While most of the proteins were in the bottom and middle fraction, significant amount of Slam1 and TbpB were found only in the liposomes fraction collected from the top layer."> Figure 4d is not fully convincing. If there is no Slam1 in liposomes (empty), Tbp or Tbp/Skp proteins should be separated from the liposomes and stay at the bottom. But, there is no Tbp detected at the bottom.

We have added more explanation line 306-308 –“ There was no trace of TbpB in both of the negative controls which could be attributed to the presence of trace amounts of proteinase K during the 18 hour ultracentrifuge run.”

8. Line 277: An explanation is needed for the B16B6 Δskp strain. It popped up out of nowhere.

We have added more explanation as recommended, now line 314-317. “To examine the role of Skp in the Slam-dependent translocation of SLPs in B16B6 N. meningitidis that contains endogenous TbpB and Slam1, we deleted the gene skp (Δskp strain) and examined its effect on surface display of TbpB (Figure 5 —figure supplement 1 and 2).”

9. Skp is known as a highly basic protein and electrostatic interaction are known to be important for binding of clients. It should be discussed whether the sequence of TbpB has any local or global sequence features for interacting with Skp.

We are currently studying this interaction. We have some preliminary data from a bioinformatic analysis though we cannot yet speculate on the specific interaction sites between TbpB and Skp in this paper.

10. Lines 322-323: "Upon their insertion into the inner leaflet of the outer membrane, LolA:SLP:Skp complexes are recognized by Slam for translocation to the cell surface."> There seems no evidence regarding whether the LolA:SLP:Skp complexes bind the outer membrane first and then are recognized by Slam. They could be directly recruited by Slam.

The reviewer is correct, we do not have that evidence. The sentence has been edited – now line 387-388. “Upon being released to the periplasm, SLP complexes are recognized by Slam for translocation to the cell surface.”

11. Lines 330-332: "Given that Slam translocon seems to require no energy input, we speculate that the folding of SLPs on the surface might provide driving force to further pull the SLPs through Slam barrel domain."> This claim (no energy input is required) may be toned down because the reconstitution system always yields low translocation efficiency compare to the cell-based system.

We agree with the reviewer and have changed the wording of this sentence- new line number 395-997 “Given that the Slam translocon does not require ATP as an energy source, we speculate that the folding of SLPs on the surface might provide driving force to further pull the SLPs through Slam barrel domain.”

12. Misleading figure calling12a) Lines 195-199: "To further validate the interaction between Skp and SLPs, a reciprocal pulldown assay was performed in which a purified His-tagged chaperone was added into the spheroplast prior to the secretion of SLPs. In this assay, we also examined whether Skp interactswith other SLPs such as hemoglobin-haptoglobin utilization protein (HpuA) – a substrate of Slam2 homolog in N. meningitidis (Hooda et al., 2016) (Figure 3b)."> This sentence should designate Figure 3c/d not Figure 3b.

Thank you, we have edited as recommended – now line 244-248.

12b) Lines 210-213: "The presence of TbpB on the surface of E. coli was detected using rabbit α-flag antibody, followed by phycoerythrin-conjugated α-rabbit IgG which fluoresces at 575nm. The results showed that only E. coli K12 Δskp mutant had significant reduction of TbpB's surface exposure (50%) compared to wildtype cells."> Figure calling should be done after this sentence.

Thank you, we have edited as recommended – now line 261-263.

12c) Line 253: "this ratio is consistent with the previous result if accounting for the 2-fold protection coming from the Skp-TbpB interaction."> "The previous result" should be specified (Figure number? Reference?)

Thank you, we have edited figure 4c as recommended – now line 304-306.

13. Missing Figure numbering: "Figure 2"> There are legends for Figure 2c but the actual Figure 2c does not exist in the figure panel.

Thank you, we have edited as recommended Figure 2 has been edited and panel c has been added.

Reviewer #2 (Recommendations for the authors):Please see my comments in the Public Review about demonstrating that the Bam complex they have purified and include in proteoliposomes is functional.The legends and labelling of figures should be revised to clarify assays and results.For instance in Figure 1 SM2 beads/DDM are not explained in the results or the legend, the numbers under the western blots are not explained, and the Y axis in panel C gives % TbpB insertion while the legend describes protection.

We have edited as recommended – an explanation for SM2 beads and DDM has been added to the results (line 124-127, and in the figure 1 legend). The number under the western blot of figure 1 is now explained in the figure legend and the % TbpB insertion has also been explained in figure legend.

Reviewer #3 (Recommendations for the authors):Although I agree that the results presented in the current manuscript support the idea that Slam1 mediates the translocation of TpbP, I think that the authors do not have enough data to make that claim. For instance, how can they rule out that binding of TpbB to Slam1 does not protect TbpB from PK digestion?

We have performed a PK digestion assay with purified Slam in DDM with TbpB – illustrating that the PK digested all the TbpB and that Slam does not provide a protective function on its own. The data was now included in the paper – Figure 1 —figure supplement 7 and discussion in main text line 133-136.

They should 1) show the time-dependence of the translocation process; 2) identify a Slam1 mutant that does not function in vivo and show that when used in the in vitro assay, it does not function either; 3) obtain evidence that TpbP is folded once inside the proteoliposomes.Thus, additional data is required to make a fully convincing case.

1) We have performed the time-dependent translocation assay and included the data in Figure 2d,e and description in main text from line 211-224

2) We have not yet done this experiment as the atomic resolution structure of Slam is still currently being resolved and more structural information is needed to make the site-directed mutations.

3) We have performed this functionality experiment for translocated TbpB. New data is included in supplementary figure 10b and detailed description from line 197-207.

The following points should also be addressed:-an unfolded protein other than TpbB should be used as a control in the translocation assay

We have performed the suggested control experiment as recommended. We denatured a periplasmic protein, named AfuA – a glucose-6-P binding protein in the periplasm, in 8M urea and performed the liposomes translocation assay. Data has been included in Figure 1 —figure supplement 7 lower panel and detailed description in text from line 136-140.

-determine whether lipidation of TpbB is required for translocation

We have data to show that the lipidation is not required for Slam translocation. However, this data is part of another manuscript in preparation that focuses on Slam specificity. If this is critical, we could include this data herein, but do not believe it influences the message of Slam functioning as a translocon with the holdase skp.

-Sup. Figure 2: a permeabilization control should be included: what if Slam1 overexpression changes the permeability of the OM

We agree that this may not have been clearly explained in this manuscript. In the new Figure (Figure 1 —figure supplement 2), we have included an Nme Slam2 + TbpB and Ngo Slam2 + TbpB experiment. Slam2 is a homolog of Slam1 and our previous published data (Hooda et al. Nature Micro, 2016) illustrated that Slam2 does not translocate TbpB (Slam2 translocate HpuA). Therefore, in this figure, Slam2 can also be considered as a negative control and permeabilization control for TbpB translocation to the surface. Thus, Slams do not function via destabilization of the membrane.